# Impacts of Changing Hydrology on Permanent Gully Growth: Experimental Results

Stephanie S. Day[1], Karen B. Gran[2], Chris Paola[3]

[1]Department of Geosciences, North Dakota State University, Fargo, 58102, USA
[2] Department of Earth and Environmental Sciences, University of Minnesota Duluth, Duluth, 55812, USA
[3] Department of Earth Sciences, University of Minnesota, Minneapolis, 55455, USA

*Correspondence to*: Stephanie S. Day (stephanie.day@ndsu.edu)

**Abstract.** Permanent gullies grow through head cut propagation in response to overland flow coupled with incision and widening in the channel bottom leading to hillslope failures. Altered hydrology can impact the rate at which permanent gullies grow by changing head-cut propagation, channel incision, and channel widening rates. Using a set of small physical experiments, we tested how changing overland flow rates and flow volumes alter the total volume of erosion and resulting gully morphology. Permanent gullies were modelled as both detachment-limited and transport-limited systems, using two different substrates with varying cohesion. In both cases, the erosion rate varied linearly with water discharge, such that the volume of sediment eroded was a function not of flow rate, but of total water volume. This implies that efforts to reduce peak flow rates alone without addressing flow volumes entering gully systems may not reduce erosion. The documented response in these experiments is not typical when compared to larger pre-existing channels where higher flow rates result in greater erosion through non-linear relationships between water discharge and sediment discharge. Permanent gullies do not respond like pre-existing channels because channel slope remains a free parameter and can adjust relatively quickly in response to changing flows.

## 1 Introduction

Permanent gullies are first-order, deeply-incised, ephemeral streams with steep head cuts. Also known as ravines (Poesen et al. 2003), permanent gullies form part of a continuum between ephemeral gullies and perennial channels. Permanent gullies can be highly erosive, accounting for 10-94% of the total sediment yield in a watershed (Poesen et al., 2003, and references therein). Gullies pose a hazard to infrastructure, agricultural productivity, and human safety through rapid propagation of head-cuts in both rural and urban environments; and they contribute to the rapid transport of sediment and nutrients from the uplands to mainstem channels (Bull and Kirkby, 2002; Gran et al., 2011; Poesen et al., 2003).

Permanent gullies initiate in places where concentrated flow can erode and move sediment (Mosley 1974; Merritt 1984; Bennett et al., 2000; Bryan, 1990; Knapen and Poesen 2010). These topographic lows may be subtle, but where they connect

with incised river valleys, steep knickpoints can rapidly form and propagate as head cuts. Gully head cut propagation occurs primarily by groundwater sapping or piping, overland flow, or a combination of these processes. Groundwater sapping typically forms amphitheatre-type headcuts (Hinds, 1925; Howard, 1988; Lamb et al., 2006) which propagate as the soil is weakened through saturation at the water table causing failure in the sediment above (Dunne, 1990). Where groundwater follows concentrated flow paths, piping can trigger head cut initiation and drive head cut retreat (Fox and Wilson, 2010; Nichols et al., 2017; Wilson, 2009; 2011). Overland flow erosion can also cause head cut propagation in multiple ways including granular erosion and slab failure. Slab failure relies on the constant wetting and drying experienced at the headcut between events. Tension cracks form as the soil dries, and during wet times water flows into these cracks dislodging the slab (Dietrich and Dunne 1993; Istanbulluoglu et al., 2005). Another form of slab failure occurs in layered substrate where a lower highly erosive layer is scoured away leaving the upper resistant layer overhanging until it eventually breaks away (Chu-Agor et al., 2008; Holland and Pickup, 1976; Robinson and Hanson, 1994; Stein and La Tray, 2002). In both of these slab failure mechanisms, head cut retreat temporarily ceases after slab failure with the slab protecting the head cut until the slab itself is eroded away (Robinson and Hanson, 1994; Istanbulluoglu et al., 2005). Granular erosion is driven by high shear stress at steep head cut slopes and can be enhanced through focused erosion of plunge pools and impinging jet scour (Alonso et al., 2002; Flores-Cervantes et al., 2006). The focus in this paper is on permanent gullies that grow as a result of granular sediment transport in a homogeneous substrate driven primarily by overland flow.

Permanent gully initiation and growth have been found to be a function of sediment texture and erodibility, slope, land-use, vegetation cover, soil moisture, headcut size, climate, and hydrology (Ambers et al., 2006; Chiverrell et al., 2007; Flores-Cervantes et al., 2006; Istanbulluoglu et al., 2005; Knapen et al., 2007; Nachtergaele et al., 2001; Poesen et al., 2003, 2011; Stankoviansky, 2003; Vandekerckhove et al., 2000; Vanwalleghem et al., 2003, 2005), but understanding the relative importance of each factor remains a challenge. Of the many variables impacting gully erosion, the two humans most readily impact are hydrology and land cover. Here the focus is on gully response to changes in hydrology.

Hydrology is altered in agricultural areas through complex changes to the volume and/or rate at which water enters permanent gully networks as overland flow, groundwater flow, or pipe flow. Agricultural development can involve direct modification of hydrology in the form of irrigation in dry climates and sandy soils or artificial drainage in wetter climates and less permeable soils. While irrigation has little impact on overland flow due to transpiration (Haddeland et al., 2005), combined changes to land cover and drainage associated with conversion from native vegetation to row-crop agriculture tend to increase the rate and volume of water entering a stream network (Blann et al., 2009; Robinson and Rycroft, 1999; van der Ploeg et al., 1999; Wiskow and van der Ploeg, 2003). Following harvest, fields are left bare for a portion of the year (1-6 months), which can lower evapotranspiration and increase volume of overland flow, particularly if the ground is frozen and infiltration cannot occur (Pierson et al., 2007). In addition, the removal of roughness elements when fields are bare can increase rates of overland

flow entering gullies (Abrahams and Parsons, 1991; Einstein and Barbarossa 1951; Eitel et al., 2011; Farres, 1978; Römkens and Wang, 1987).

Although changes to hydrology in urban areas are different from those in agricultural areas, the impacts on permanent gully hydrology can be similar, with increased impervious surface area and rerouting of water through infrastructure leading to more
direct hydrologic connections between uplands and gullies. Although the presence of storm sewers and water retention ponds may mitigate the effect of increased impervious area, many urban areas see an increase in overland flow rates and the overall volume of overland flow (Arnold and Gibbons, 1996; Espey et al., 1965; Paul and Meyer, 2001; Seaburn 1969). Increased gullying is particularly of concern in urban areas with rapid population growth, intense rainfall, and infrastructure that promotes flow concentration into erodible areas (Adediji et al., 2013; Ebisemiju, 1989).

Permanent gullies evolve over shorter time scales than larger pre-existing channels. Previous numerical and physical experiments of gully evolution identified two stages to gully growth: an early stage of rapid evolution and a later stage of growth within a more static form (Parker, 1977; Kosov et al., 1978; Parker and Schumm, 1982; Sidorchuk, 1999; Bennett et al., 2000). During this early phase, gullies should be more responsive to changes in hydrology as both channel cross-sectional geometry and longitudinal profile geometry are more adaptable. During the latter phase, changes in hydrology would be
accommodated through erosion focused on head-cuts at the channel tips and channel incision and widening with ensuing adjustments to side slopes (Sidorchuk, 1999; 2006).

Because anthropogenic alterations to the landscape involve both changes in flow rate and volume, we chose to investigate those two aspects of hydrologic change using a physical experimental basin. Two different experimental substrates were used to measure the effects of changing flow rates on both detachment-limited and transport-limited systems. These two types of
erosion represent a continuum, where the volume of sediment carried out of the system is controlled by the ability of the flow to dislodge sediment from the substrate vs. the ability of the flow to carry easily eroded sediment. Commonly "detachment-limited" is used to describe bedrock rivers, while "transport-limited" is used to describe alluvial rivers. Given that permanent gullies are incisional systems and may be down-cutting through non-alluvial substrate, the detachment-limited model may be more appropriate for permanent gully systems than classic transport-limited models for alluvial systems.

Physical experiments, like the ones we discuss here, offer a setting where most variables can be controlled and time scales for channel evolution are greatly reduced. Moreover, using experiments allows us to make measurements at a high spatial and temporal density to ensure that much of the variability in the system is captured. Such experiments are not intended as scale models; rather, they are small systems in which scale-independent processes can be studied under controlled conditions (Paola et al., 2010). As such the experiments presented here add to the body of literature focused on channel formation across a
continuum of scales from rills over short time scales to head-cutting rivers over much longer time scales (i.e. Stein and LaTray,

2002; Grovers et al., 2007). The discussion of this work focuses on permanent gullies or ravines because these features tend to form over relatively short timescales geologically, but similar timescales over which pre-existing channel networks might also evolve, thus allowing for a comparison between the two. Permanent gullies can also cause deep and rapid incision, potentially having a significant effect on infrastructure. Here we utilize physical experiments of permanent gullies to focus on a basic question: how does changing the flow rate of a fixed quantity of water change the total volume of sediment removed and the morphology of the resulting permanent gully?

## 2 Methods

We performed experiments at the St. Anthony Falls Laboratory at the University of Minnesota, in Minneapolis, Minnesota. These experiments tested how different overland flow rates affect erosion and permanent gully growth on a series of permanent gullies initiated through base level fall, a common feature of post-glacial landscapes in the Upper Midwest, USA (Belmont et al., 2011; Gran et al., 2013; Lenhart et al., 2013). We represented a range of natural permanent gullies using two different substrates. These substrates varied in material and grain size with one representing a more cohesive detachment-limited system and the other a transport-limited system. The fine-grained substrate was composed of 12 μm ceramic spheres (Zeeospheres), with a density of 2.5 g cm$^{-3}$ (Zeeospheres G series data sheet). The natural cohesion of this substrate caused it to behave as a detachment-limited system, as empirically shown in the results section of this paper. The coarser substrate was composed ofquartz sand with a median grain size of 96 μm and a density of 2.65 g cm$^{-3}$ (AGSCO technical data). This sand lacked cohesion and behaved as a transport-limited system. The experimental basin size varied for each material type; the basin used with the fine-grained substrate was 1 x 1 m, while the basin filled with the sand substrate was 2 x 4 m thus allowing room for the more rapid growth of the permanent gullies in the more easily erodible substrate. For both substrates, water flowed out through a 0.076 m wide notch at the downstream end of the basin (Fig. 1). To initiate each run, we dropped the notch to 0.14 m below the surface of the substrate, thus creating a single abrupt base level drop. During each run a knickpoint developed at this notch and propagated upstream, carving a deep channel in the substrate.

To initialize each run, the substrate material was mixed with water and smoothed into the basin to create a flat bed. Experiments began with a fully saturated bed to ensure that water flowed over the surface rather than infiltrating into it. We determined that the bed was at saturation and no longer over-saturated when there was no longer water on the surface of the sediment and water was no longer flowing out the downstream outlet. We analysed the bed saturation by visual inspection rather than waiting a standard length of time because changes in humidity and temperature affected evaporation rates. Beginning with a saturated

substrate allowed us to assume that the net flow was relatively constant in the cross-channel direction and no water was lost to infiltration into dry sediment.

For each experimental run, water flowed over the uniform substrate as overland flow. We generated an even sheet flow by allowing water to flow over a broad crested weir as water was added to an upstream settling basin (Fig. 1). We controlled the flow rate of the water entering the settling basin with a constant-head tank, which received water from a tank with a predetermined volume of water (190 or 380 litres). The flow rate was held constant through the entire run when only 190 litres of water were used. When 380 litres of water were used, the flow rate was held constant for the first 190 litres, then increased and held constant for the second 190 litres. Flow rates varied from 4 to 311 ml $s^{-1}$ (4 x $10^{-6}$ to 311 x $10^{-6}$ $m^3$ $s^{-1}$)in the fine-grained substrate and 55 to 262 ml $s^{-1}$ (55 x $10^{-6}$ to 262 x $10^{-6}$ $m^3$ $s^{-1}$)in the sand (Table 1; Fig. 2).

We collected topographic data before, during, and after each experimental run using a fully automated topographic scanner developed by engineers at the St. Anthony Falls Laboratory. The topographic scanner uses a standard laser that collects 2000 points per second in a gridded pattern. Data were collected with 20 x 25 mm point spacing and approximately 0.5 mm vertical resolution. For each experiment, we collected 2-5 topographic scans per 190 litres of water. The experiment was paused at these intervals to complete the scans; ensuring that no data were lost in areas where water was present.

Topographic data were gridded to form a Digital Elevation Model (DEM). To determine the total volume of sediment removed during each experimental run, we subtracted the DEM of the last scan from the DEM created from the initial scan over the flat initial surface. At each cell the vertical change was multiplied by the area of the cell to give a volume change. The total volume change was summed for each experimental run. We also calculated the volume of sediment eroded at intermediate scans to measure sediment flux over time. Because these data only provide coarse temporal measurements of sediment flux, we sampled sediment flux out of the experiment for 5 seconds every minute during experimental run F-11 to measure sediment flux at higher time resolution.

We imported the DEMs into ArcGIS to measure channel width and slope using the profile tool in the 3D analyst toolbox. Channel width is defined as the distance between channel banks on a cross section. Where multiple channels formed the cross section was selected just below the confluence of these channels. In the fine-grained substrate, channel width was roughly equal to the valley width along the full channel length. In the sand substrate, channels meandered, and therefore we measured channel width in the most upstream portion of the channel where migration had not yet occurred, and the valley width was equal to the channel width. We also measured the channel length and used this measurement to approximate knickpoint retreat rate. In both substrates we measured the channel length from the outlet to the break in slope at the flat upland surface. Slope was measured along the channel profile. In the fine-grained substrate we measured channel slope three separate ways: along the bed, along the knickpoint, and the average channel slope along the complete channel, including both channel bed and

knickpoint zones. Where there was more than one knickpoint, we measured the bed slope and knickpoint slope for each section and averaged. Only one slope measurement, the average channel slope, was made on channels formed in the sand substrate because knickpoints were less prominent. In both substrates, average channel slope is a function of channel length, because elevation change for each channel is the same.

## 3 Results and Analysis

These experiments show that the total eroded volume shows no consistent trend with flow rate. For all 190 litre experimental runs, the volume of sediment removed ranges from $8.9 \times 10^{-3}$ to $2.4 \times 10^{-2}$ m$^3$ with no clear trend (Fig. 3A). This result was the same for both the sand and the fine-grained substrate. Similarly, there is no trend in the volume of sediment removed in the 380 litre runs, yet the volume of eroded sediment is greater ($2.8 \times 10^{-2}$ to $3.9 \times 10^{-2}$ m$^3$) than for the 190 litre runs (Fig. 3). Put another way, the sediment discharge is linearly related to water discharge (Fig. 3B). For a natural system this implies that in a given storm event the amount of sediment eroded from a permanent gully is proportional to the amount of precipitation rather than the storm intensity assuming flow rate generates shear stresses that surpass the critical shear stress of the sediment.

In experimental run F-11, the sediment discharge peaked early and rapidly decreased (Fig.4). This pattern is also seen in several other experimental runs using the coarser resolution erosion data collected from the topographic scans, most prominently in run F-6 (Fig. 5). The experimental runs where this pattern was observed were also those where the channel interacted with a wall (see supplemental data). After a channel touches a wall it is no longer able to widen, and flow preferentially remains against that smooth surface. While it was observed that the channels appeared to propagate most rapidly early in the experiment and then slow with each successive time interval, the scan data suggest that the total volume of sediment removed was relatively constant throughout the experiment except where the channel interacted with a wall. For the 380 litre runs, where discharge was increased, there was a corresponding increase in sediment discharge, which was then maintained throughout the second half of the experiment.

### 3.1 Channel Morphology

While sediment volumes removed are roughly the same in the fine-grained and sand channels, there are differences in how they erode. Channels in the fine-grained substrate erode primarily via head cut propagation. In contrast channels in the sand substrate erode due to head cut propagation as well as lateral channel migration (see supplementary data).

The experiments also demonstrate a relationship between channel width and discharge. Generally, higher flows resulted in shorter, wider channels (Table 2). In particular, the 380 litre runs reveal both the importance the total flow volume has on erosion volumes and how increasing flow rate affects channel geometry. In the cohesive fine-grained substrate, higher flows in the second half of the experiment formed a wider channel upstream of the already eroded gully, without altering the pre-

existing channel during the rest of the experiment (Fig. 6). Channel widths before the increased flow ranged from 0.16 to 0.40 m for a flow rate of 73 x $10^{-6}$ $m^3$ $s^{-1}$ to 77 x $10^{-6}$ $m^3$ $s^{-1}$ and increased to 0.28 to 0.61 m for flow rates ranging from 143 x $10^{-6}$ $m^3$ $s^{-1}$ to 234 x $10^{-6}$ $m^3$ $s^{-1}$. The channels formed in the sand substrate responded differently; these channels widened along the entire channel length when flow was increased (Fig. 6). In the sand channels widths varied from 0.14 to 0.19 m for initial flows

which varied from 42 x $10^{-6}$ $m^3$ $s^{-1}$ to 81 x $10^{-6}$ $m^3$ $s^{-1}$, and the entire channel width increased to 0.24 to 0.43 m for the higher flows ranging from 185 x $10^{-6}$ $m^3$ $s^{-1}$ to 225 x $10^{-6}$ $m^3$ $s^{-1}$ (Table 2).

The hydraulic geometry equation for width developed by Leopold and Maddock (1953) was used to quantify the relationship between width, $W$, and discharge, $Q$, for these experiments to determine how the experimental channels compare with natural channels:

$W = aQ^b$                                                                 (1)

where $a$ and $b$ are constants. The range of empirically-derived exponents for $b$ is between 0.3 and 0.5 derived from field measurements of both bedrock and alluvial streams (Knighton, 1998; Leopold and Maddock, 1953; Montgomery and Gran, 2001; Whipple, 2004). Studies of rills and ephemeral gullies have found that these relationships hold with a $b$ value of 0.4 – 0.5 (Natchergaele et al., 2002; Torri et al., 2006).

To test how well these field-measured values describe our experimental channels, we calculated width using the reported values for $b$ and iterated to find the best fit for $a$. In addition, we also iterated to determine what $b$ value most accurately describes the trend in our experimental data. The root mean square error (RMSE) was calculated for each data set and is reported as a percentage of the average measured width value.

Using the field-derived $b$ exponents ranging from 0.3-0.5, the RMSE is 29-31% of the average measured width in the fine-
grained substrate. The best fit for the experimental data results in a b value of 0.27 resulting in an RMSE of 28.5% (Fig. 7). In the sand substrate the best fit for the data falls within the range of field-derived exponents. The error is minimized when $b$ is 0.39 resulting in an RMSE of 9% (Fig. 7).

### 3.2 Modelled Sediment Transport

Results from these experiments appear to conflict with standard sediment transport equations which generally predict a non-
linear increase in sediment flux as discharge increases (e.g. Engelund and Hansen, 1967; Meyer-Peter and Müller, 1948; Parker, 1990; Wilcock and Crowe, 2003), yet these equations typically include an additional factor, often slope, which, if also varied, can account for changes in discharge resulting in a linear increase in sediment flux. Below are examples of commonly

used sediment transport models that, when applied to these experiments, account for the linear nature of the sediment flux discharge relationship.

Commonly erosion in detachment-limited systems is modelled by the stream power incision model (Howard and Kerby, 1983):

$dz/dt = kQ^{(m_d)} S^{(n_d)}$                                                        (2)

where $k$, $m_d$ and $n_d$ are constants, $dz\,dt^{-1}$ is vertical erosion rate, and $S$ is the slope. In the detachment-limited fine-grained substrate, all erosion took place at the knickpoint and therefore we consider only knickpoint retreat rate and use $S_k$, knickpoint slope, in place of $S$. Because this rate is a horizontal retreat rate rather than vertical incision rate we must convert Eq. (2) appropriately:

$U{\cdot}S_k = dz/dt$                                                                          (3)

$U = kQ^{(m_d)} S_k^{(n_d-1)}$                                                                     (4)

where $U$ is the knickpoint retreat rate. The exponents $m_d$ and $n_d$ have been derived for a variety of natural environments. Typically the $m_d\,n_d^{-1}$ ratio (concavity index) is approximately 0.35 - 0.6 (Whipple and Tucker, 1999; Baldwin et al., 2003). The exponent $n_d$ ranges between 2/3 and 5/3 depending on the erodibility of the substrate where more easily eroded sediment
has a lower value for the exponent $n_d$ (Foley 1980; Howard and Kirby 1983; Whipple et al., 2000). The exponents $m_d$ and $n_d$ for uniform detachment-limited landscapes (i.e. badlands) reported by Howard and Kerby (1983) are 4/9 and 2/3, respectively. While the $m_d\,n_d^{-1}$ ratio is slightly higher than reported elsewhere, we used these values to model erosion in our experiments because, like the badlands, our experimental set-up was spatially homogeneous and easily eroded. RMSE was measured between the calculated $U$ and the measured $U$ and minimized by modifying the coefficient $k$. The RMSE was 25% of the
measured average knickpoint retreat rate for the fine-grained substrate. The detachment-limited equation is not appropriate for the sand substrate because there is both headward and lateral erosion, which is not captured by the equation. In addition, the knickpoint slope cannot be measured in the sand substrate.

Sediment loads from the sand substrate were modelled with the transport-limited equation (Pelletier, 2011):

$q_s = kQ^{(m_t)} S^{(n_t)}$                                                                           (5)

where $k$, $m_t$ and $n_t$ are constants and $q_s$ is the volumetric unit sediment flux. This equation can be derived from the Engelund and Hansen (1967) equation where both $m_t$ and $n_t$ equal 5/3. Engelund and Hansen (1967) is the ideal equation to use because it does not have an incipient motion threshold as many other sediment transport equations do. Here again the RMSE was

measured between the measured and calculated $q_s$ and minimized using the coefficient $k$. The RMSE was 41% of the average volumetric unit sediment discharge for the sand substrate. The transport-limited equation was also tested with the fine-grained substrate. For this test the average slope was used for $S$ and the RMSE calculated was 86% of the average volumetric unit sediment discharge. This high RMSE value supports the observation that the fine-grained substrate does not behave as a transport limited system.

## 4 Discussion

Our experiments demonstrate that, over a range of conditions, the sediment volume eroded during permanent gully growth under application of a fixed volume of water is independent of the rate at which the water is supplied. Thus, sediment discharge is linearly related to water discharge in both detachment-limited and transport-limited systems. These results contrast with data from many pre-existing streams where changing flow intensity has resulted in increased erosion volume (i.e. Boateng et al., 2012; Ma et al., 2010; Naik and Jay, 2011), yet these data may not be directly applicable to early-stage gullies. We suggest that because gullies are actively evolving in response to a given hydrology, the channel morphology that develops reflects that hydrology, with erosion balanced by altering channel slope. This is supported by sediment transport Eqs. (4) and (5) which, when applied to our experiments, predict the measured sediment discharges by including the effects of both discharge and slope. In pre-existing channels where channel slope may take tens of thousands of years to adjust to changing flows, both equations would predict a non-linear increase in sediment discharge with increasing water discharge. Gullies evolve more rapidly in response to the imposed discharge and can balance erosion by adjusting channel slope in response to a change in the hydrologic regime.

Moreover, our findings suggest that anthropogenic changes to discharge regime could affect channel morphology (i.e. channel width), without changing sediment output derived from permanent gullies. Where water discharge was increased in the 380 litre runs, the channel quickly evolved in response to the new discharge by creating a wider channel. In these experiments the observed response to the increased discharge differed for the fine-grained and sand substrate (Fig. 6) suggesting cohesion may be an important factor in how a channel initially responds to a new discharge regime.

The results of our experiments follow the hydraulic geometry relationship for width in Eq. (1) although with lower exponents than usually measured in field studies for alluvial and bedrock channels (Knighton, 1998; Leopold and Maddock, 1953; Montgomery and Gran, 2001; Whipple, 2004) and rills and ephemeral gullies (Nachtergaele et al., 2002; Torri et al., 2006). While this relationship is typically applied to describing width or discharge changes in a single channel, it works here for these separate channels because each comparable channel is carving through the same substrate. In the fine-grained substrate the empirical exponent $b$ for these data is lower than has been derived for natural channels. This may be a result of the steep channel walls developed in these experiments; in natural permanent gullies where near vertical channel walls are less common,

we expect that this exponent would be closer to the reported values. In the sand substrate where steep channel walls could not develop, the empirical exponent $b$ is 0.39 which is only slightly lower than the range typically considered for alluvial channels (Rodriquez-Iturbe and Rinaldo, 1997), but similar to exponents found in rills and ephemeral gullies (Nachtergaele et al., 2002; Torri et al., 2006).

Channel width has also been modelled by Wells et al. (2013) who found that both slope and discharge play a role in setting channel width. This relationship was tested for the results of these experiments, yet it was not as strong as the relationship with discharge alone. Slope in the Wells et al. (2013) study could not change during the experiment, which contrasts with our experiments where slope is a free parameter that adjusts to discharge. This finding further supports the distinction between rapidly evolving channels like permanent gullies, and more stable systems where slope does not change as quickly causing

channel adjustments to occur primarily through changes in channel width and through a non-linear erosion response.

Experiments focused on headcut growth completed by Bennett et al. (2000) also reported a linear relationship between water discharge and sediment discharge, yet the water discharge was lower, and the slope of the relationship was much higher in our results. This may suggest that a nonlinear relationship between sediment and water may develop over a wider range of flow rates than tested here, yet more research would be required. In addition, Bennett et al. (2000) noted two dominant processes

for head cut migration: surface seal failure, which is similar to slab failure reported in other papers, and plunge pool scour, where headcut migration is driven by undercutting. Although both of these mechanisms could lead to large blocks of sediment collecting at the base of the headcut, creating periods of quiescence in headcut migration, the authors do not indicate that headcut migration stalled in their experiments. The erosion mechanisms described by Bennett et al. (2000) are similar to the mechanisms observed in our experiments, in that there was a continuous headcut propagation, although we did not observe

significant plunge pool development.

In a numerical study, Istanbulluoglu et al. (2005) tested the effect of changing rain intensity while storm volume was held constant. The modelled results of their study showed an increase in the volume of sediment eroded as intensity increased. This result is in direct conflict with our results, yet there are many distinctions between the two studies that may explain this discrepancy. The Istanbulluoglu et al. (2005) model assumed gully erosion due to slab failure in a detachment-limited system.

While the fine-grained substrate in our physical experiments was also detachment limited, erosion occurred grain by grain rather than as large blocks. Once these grains were detached, the flow was easily able to carry them through the channel and there was no measurable deposition in any of the experimental runs. In contrast, erosion in the slab failure model occurred in response to pore pressure build up in tension cracks resulting in large failures. This slab failure model does not require that the

flow be able to carry the detached sediment and often resulted in deposition at the toe of the knickpoint, which increases resistance to future failure.

It is likely that both slab failure and granular knickpoint propagation occur in permanent gullies throughout the world. The relative importance of each process is dependent on the substrate and the knickpoint slope. Tension cracks develop behind steep slopes where shrinkage occurs due to desiccation and horizontal tensile stresses generated in large part by gravity are greater than the tensile strength of the sediment (Darby and Thorne, 1994). Cracks like this are likely to form on the landscape above steep head cuts in cohesive sediment, between storm events. In neither our study nor the Bennett et al. (2000) did we model individual storm events rather than a constant overland flow. If we had, it is likely we would have also developed tension cracks in the cohesive substrates. Because this was outside of the scope of these experiments it is difficult to form accurate conclusions on how the development of tension cracks and the subsequent failure events would have altered these results, but an analogous study that encouraged erosion by slab failure would be a useful extension.

Our experimental results are not clear with regard to Sidorchuk's (1999) two-stage gully evolution model. For most runs it appears that the second stage of gully evolution was not achieved. In a few specific cases, most notably, runs F-6 and F-11 there does appear to be an early peak and later decrease and stabilization in sediment discharge. Surprisingly the results of these runs were not among the lowest total sediment discharges, as might be expected where the second stage was achieved. It is possible that if all the runs were allowed to continue we may have reached a stable system where the second stage of gully evolution was achieved. If this was allowed to occur, we would anticipate that additional water at the same discharge would not cause measurable erosion, potentially altering the relationship we have observed. What is not clear is how long it takes for this steady state to occur, and how this may relate to discharge.

Based on the results of this study and the comparison with previous gully studies, it is important to consider a wide range of variables when mitigating permanent gully erosion. The results from our experiments suggest that during early stages of permanent gully growth, increasing overland flow rates will not result in increased sediment yield, if the volumes of water delivered are not changed. Moreover, while sediment loads are not affected by changing flow rates, channel morphology is. Another important variable to consider is the mode of head cut retreat. The experimental results apply in environments with steep slopes where erosion is grain-by-grain. In places where tension cracks develop, the slab failure mechanism highlighted by Istanbulluoglu et al. (2005) might dominate. If the tension crack failure mechanism is the dominant process of head cut

retreat, flow rates and storm intensity may become more important than they were in our study because the slabs may require a higher threshold to break up and mobilize, allowing further head-cut propagation.

## 5 Conclusion

These experiments highlight how young incising channels like permanent gullies can respond to changing hydrology differently than higher order channels that are later in their evolution. A relevant future study should investigate how natural gullies, which have a great deal more variability than this experimental system, respond to changing hydrology. The conclusions from this project are outlined below:

- The experiments here suggest that water volume, rather than discharge, controls the total volume of erosion during permanent gully formation. This result holds true for both transport-limited and detachment-limited systems.
- As long as slope is a free parameter in these rapidly-evolving systems, changes in flow rate can be accommodated through an adjustment in both cross-sectional and longitudinal channel geometry. Wider channels were typically shorter and thus steeper.
- In both substrates, variations in channel width were described by the hydraulic geometry relationship proposed by Leopold (1953), with wider channels forming in response to higher discharge.
- Sediment transport in sand and fine-grained substrates was well described by the transport-limited sediment flux equation and the detachment-limited stream power equation, respectively.

**Competing Interests**

The authors declare that they have no conflict of interest.

**Acknowledgments**

This work was funded in part by a Geological Society of America Graduate student research grant and by the National Center for Earth-surface Dynamics (NCED), which is funded by the Office of Integrative Activities, National Science Foundation grant EAR-0120914. We appreciate comments and reviews from Robert Wells, Valentin Wendling, and two anonymous reviewers.

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

30

Table 1: Experimental Run Parameters

| Run | Substrate | Water Volume (liters) | Total Time (min) | Flow 1st 190 liters ($\times 10^{-6}$ m$^3$ s$^{-1}$) | Flow 2nd 190 liters ($\times 10^{-6}$ m$^3$ s$^{-1}$) |
|---|---|---|---|---|---|
| F-1 | Fine | 190 | 60.0 | 52.58 | |
| F-2 | Fine | 190 | 32.0 | 98.58 | |
| F-3 | Fine | 190 | 16.0 | 197.16 | |
| F-4 | Fine | 190 | 21.25 | 148.45 | |
| F-5 | Fine | 190 | 11.0 | 286.77 | |
| F-6 | Fine | 190 | 872.0 | 3.62 | |
| F-7 | Fine | 190 | 10.13 | 311.40 | |
| F-8 | Fine | 190 | 21.5 | 146.72 | |
| F-9 | Fine | 190 | 15.5 | 203.52 | |
| F-10 | Fine | 380 | 91.5 | 71.69 | 66.41 |
| F-11 | Fine | 380 | 65.0 | 73.36 | 143.39 |
| F-12 | Fine | 380 | 54.5 | 76.94 | 233.67 |
| S-13 | Sand | 190 | 25.25 | 124.93 | |
| S-14 | Sand | 190 | 56.5 | 55.83 | |
| S-15 | Sand | 190 | 13.5 | 233.67 | |
| S-16 | Sand | 190 | 12.0 | 262.88 | |
| S-17 | Sand | 190 | 20.25 | 155.78 | |
| S-18 | Sand | 380 | 56.0 | 80.88 | 185.56 |
| S-19 | Sand | 380 | 71.5 | 55.34 | 217.55 |
| S-20 | Sand | 380 | 89.0 | 42.06 | 225.32 |

Table 2: Experimental Results

| Run | Volume Sediment Removed ($m^3$) | Average Slope | Bed Slope | Knickpoint Slope | Width 1st 190 liters (m) | Width 2nd 190 liters (m) |
|-----|-----|-----|-----|-----|-----|-----|
| F-1 | 0.011920 | 0.17 | 0.06 | 1.19 | 0.22 | |
| F-2 | 0.024506 | 0.20 | 0.04 | 0.60 | 0.48 | |
| F-3 | 0.017937 | 0.21 | 0.04 | 0.36 | 0.24 | |
| F-4 | 0.013776 | 0.27 | 0.03 | 0.62 | 0.43 | |
| F-5 | 0.010384 | 0.29 | 0.04 | 1.22 | 0.26 | |
| F-6 | 0.014903 | 0.12 | 0.10 | 0.27 | 0.13 | |
| F-7 | 0.08913 | 0.42 | 0.11 | 0.71 | 0.38 | |
| F-8 | 0.013908 | 0.37 | 0.06 | 0.67 | 0.43 | |
| F-9 | 0.018645 | 0.22 | 0.07 | 0.58 | 0.52 | |
| F-10 | 0.039053 | 0.15 | 0.05 | 0.81 | 0.40 | 0.49 |
| F-11 | 0.030158 | 0.14 | 0.05 | 0.56 | 0.16 | 0.28 |
| F-12 | 0.035532 | 0.16 | 0.03 | 0.29 | 0.38 | 0.61 |
| S-13 | 0.017762 | 0.06 | | | 0.20 | |
| S-14 | 0.018649 | 0.07 | | | 0.13 | |
| S-15 | 0.023796 | 0.06 | | | 0.28 | |
| S-16 | 0.016590 | 0.06 | | | 0.27 | |
| S-17 | 0.023569 | 0.06 | | | 0.29 | |
| S-18 | 0.031174 | 0.07 | | | 0.19 | 0.39 |
| S-19 | 0.030587 | 0.06 | | | 0.14 | 0.43 |
| S-20 | 0.028028 | 0.06 | | | 0.15 | 0.24 |

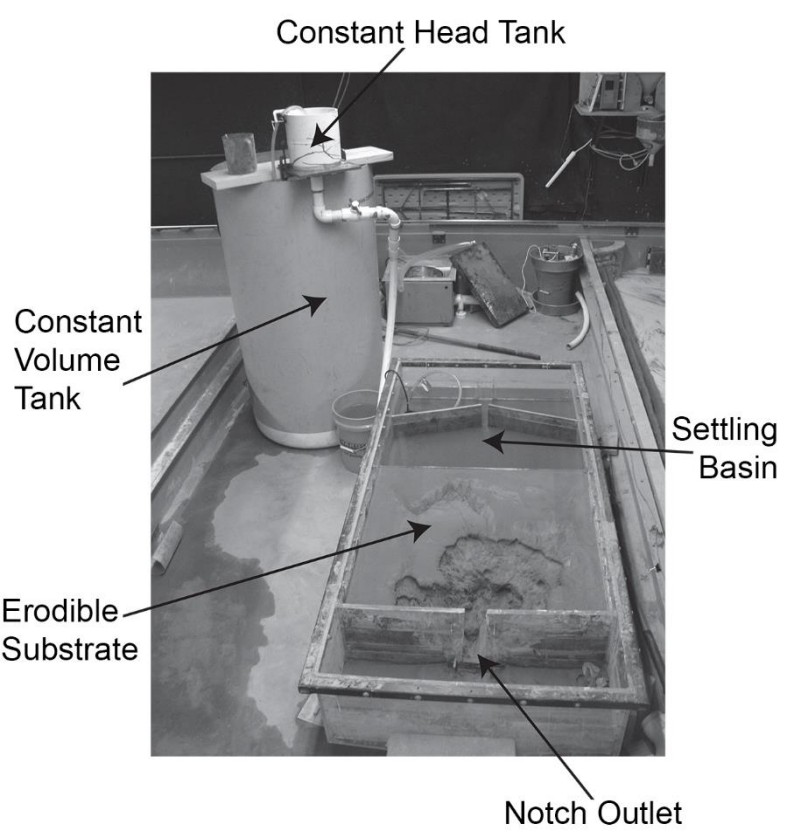

**Figure 1: The experimental set up shown here allows water to flow from a settling basin over an erodible substrate and out through a 7.6 x 14 cm notch. The flow rates entering the basin range from 4 x 10⁻⁶ to 311 x 10⁻⁶ m³ s⁻¹ and are controlled by a constant head tank. For each run a constant volume of water either 190 or 380 litres is run over the erodible substrate. This figure shows the set up for the fine-grained substrate, but the sand substrate set up was similar, yet the erodible substrate was larger.**

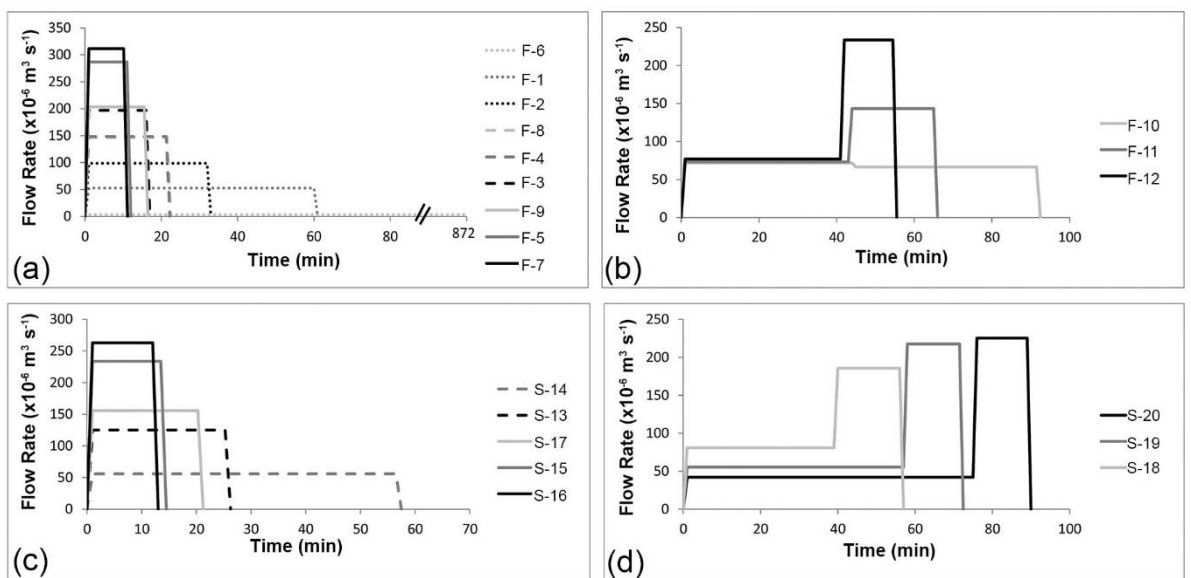

**Figure 2: These hydrographs show the range of flow rates tested. Each set of curves is for a separate set of experimental runs. A) 190 litres of water over the fine-grained substrate. B) 380 litres of water over the fine-grained substrate. C) 190 litres of water over the sand substrate. D) 380 litres of water over the sand substrate.**

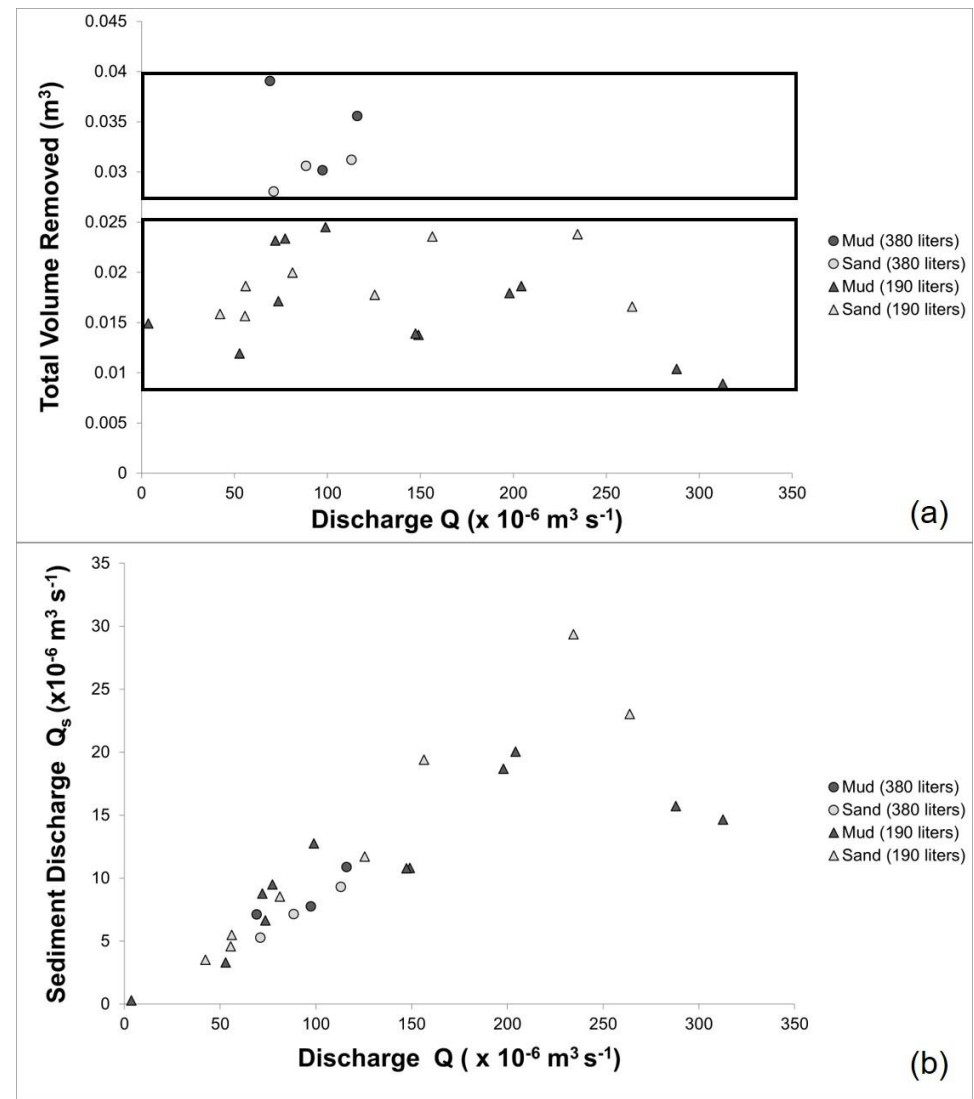

**Figure 3:** The results from these experiments show that the total volume of sediment removed is not dependent of flow rate (A) or that sediment discharge is linearly related to water discharge (B). These charts also show that while sediment discharge is the same for the 380 litre runs as it is for the 190 litre runs at a given discharge, the total volume of sediment removed is greater in the 380 litres runs.

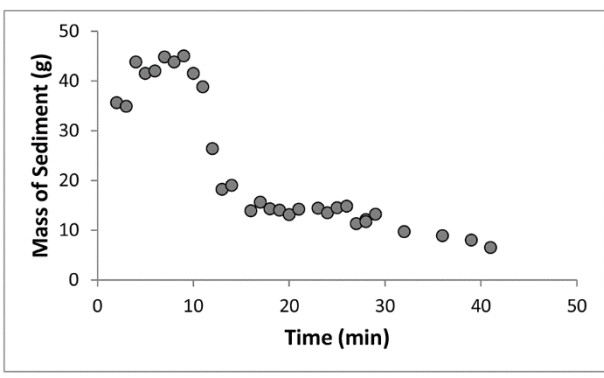

**Figure 4: Samples collected for 5 seconds every minute during the first 190 liters of run 11 show a decrease in sediment load with time after an initial peak.**

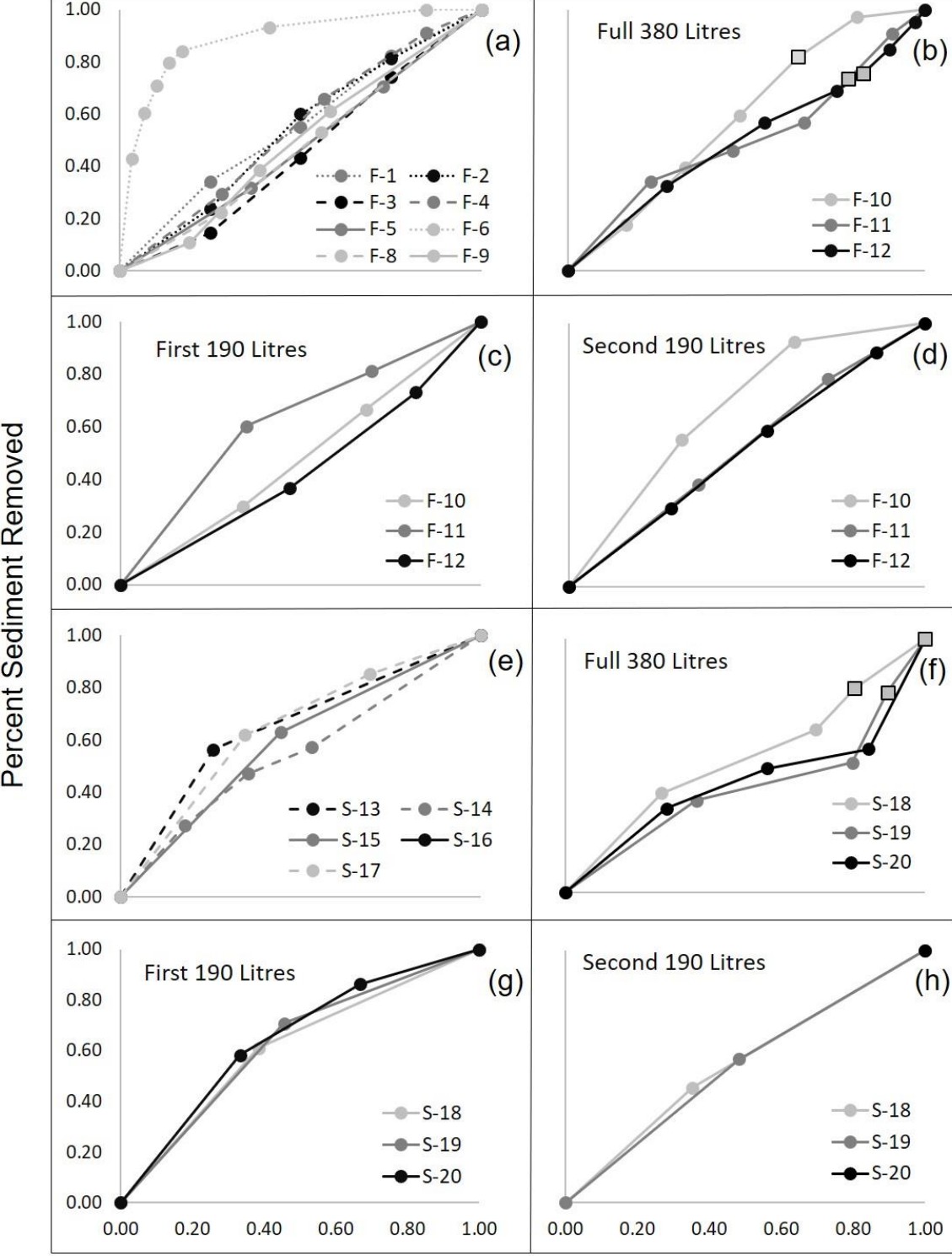

**Figure 5: In general, sediment discharge was near constant throughout the experiments as indicated by the linear relationship between percent time elapsed and percent of total sediment removed in the graphs above. This result was generally consistent between for both fine-grained substrate (a-d) and the sand substrate (e-h). Results from the 190 litre runs are graphed in a and e and the 380 litre runs are graphed in b and f with the first half broken out in c and g and the second half in f and h.**

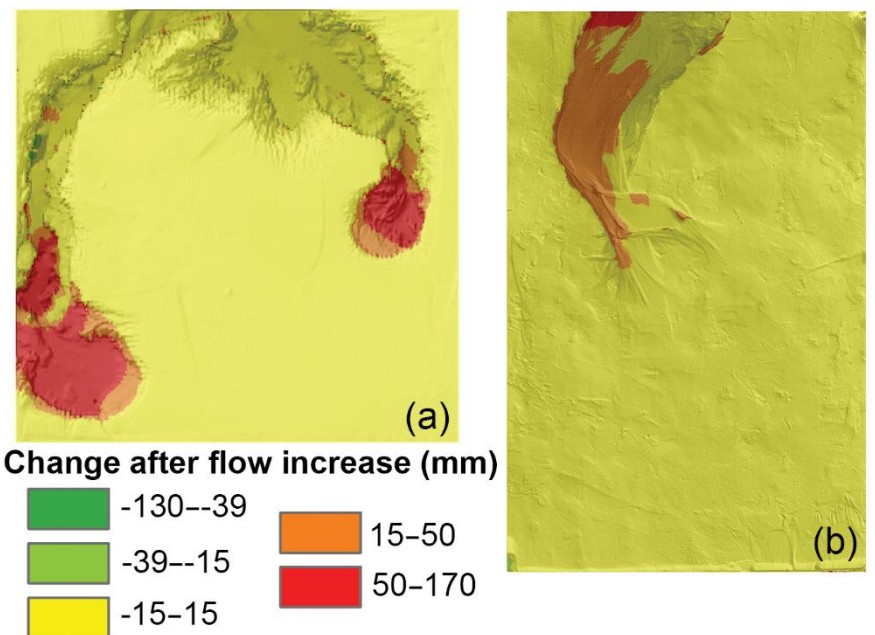

**Figure 6: These differenced DEMs show how changes in flow impact channel width. In the fine-grained runs (A) the width changed in the newly formed channel, while in the sand runs (B) the channel width was altered along the entire channel length. Flow is bottom to top in both images.**

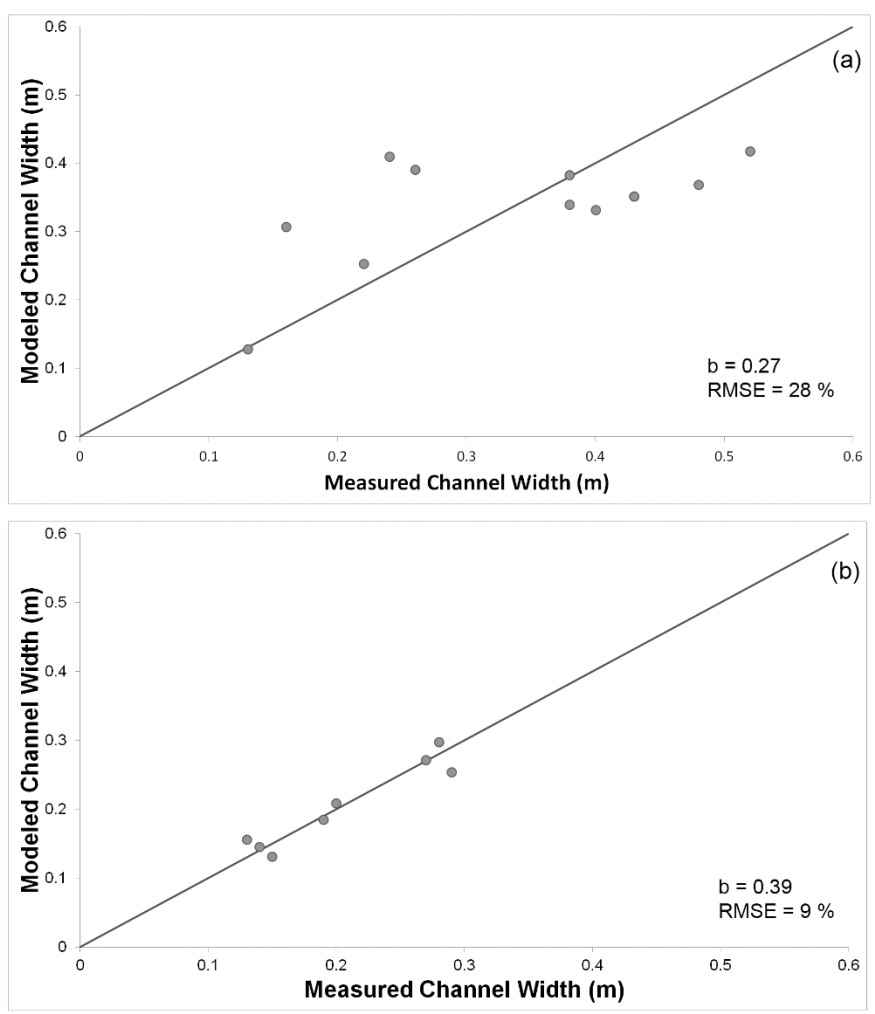

**Figure 7: Channel width can be modelled using the hydraulic geometry relationship (Eq. 1). In both plots the solid black line is a 1:1 line A) The relationship measured in the channels formed in the fine-grained substrate has a constant *b* value that is less than the values commonly observed in nature. B) The relationship measured in the channels formed in the sand substrate has a constant *b* value of 0.4, which in the intermediate range of the commonly proposed constants.**