# Peer review of "Impacts of Changing Hydrology on Permanent Gully Growth: Experimental Results"

_Hydrology and Earth System Sciences, 2017_

## Short Comment (SC1) · 2 Nov 2017

Comments on Impacts of Changing Hydrology on Ravine Growth: Experimental Results Hydrol. Earth Syst. Sci. Discuss., https://doi.org/10.5194/hess-2017-567

Dear authors, dear reviewers, dear editors,

I read the experiments with interest. I consider it as an interesting contribution to erosion science. I have a main comment to assess about the interpretation. The total sediment removal volume (Vs) is shown to be not dependent of the flow rate (discharge) (fig 5a). This implies the presented linear relation between the mean water discharge and the mean sediment discharge (fig 5b), as the total volume of water (Vw) is constant, and the total experiment duration (T) is used to calculate both sediment

discharge (Qs) and water discharge (Qw): Qs=Vs/T and Qw=Vw/T.

It is also presented that the sediment discharge is not constant during the experiment, at the beginning of each experiment the sediment discharge is high and increases to reach a maximum during the ravine formation, then decreasing to get a significantly lower value (fig 4 and 5). If the water discharge increases during the experiment, a second peak of sediment discharge is observed following the liquid discharge increase. This is clearly shone on the graphs and explained in the text.

For the longest runs (ie, low water discharge as the water volume is constant in the experiment design) the sediment flux at the end of the experiment seams to have reached a relatively constant value, drastically lower as during the ravine formation (run 6 and first stage of run 11 for example). Thus the total removal volume would roughly corresponds to the volume exported during the ravine formation, with a negligible contribution of the flux existing during a following relatively steady state. Indeed if the experiments would be conducted with a higher volume of water delivered at the same discharges, we could expect a quite unchanged removal volume.

Thus it seams confusing to present measured erosion as solid discharge (averaged during the experiment time) and it would be more clear to present it as total removal volume (both in the graph and in the text).

We could consider a first stage during which the ravine forms and adjusts to the constant discharge, with a removal volume not dependent on the discharge, and a second stage where the ravine is in steady state with this discharge. During ravine formation and adjustment to the discharge, the eroded volume would not be dependent on the discharge. During the second stage, the solid discharge is drastically lower and could follow the dependence between solid and liquid discharges known in river systems.

As a second sediment load peak is observed when the discharge is increased (in the experiment conducted with larger water volume – runs 10-12 and 18-20), the total eroded load in a ravine could be function of water discharge fluctuation and temporar-

ily. Note that the total eroded load is higher for experiments conducted with increasing water discharge (and higher water volume) (figure 3a), which could support this hypothesis. Nevertheless, in those experiments the final steady state (if existing) has not been reached (fig 5 c and 5d) mitigating any interpretation.

Considering the previous comments, I put into question the discussion about the independence between liquid discharge and sediment discharge, it also put into question part of the discussion on the implication of the presented study results for natural systems.

Finally I assess some specific comments:

As on the precise measurement, the sediment flux is not null at the beginning of the experiment (figure 4), there are no reason to link the first point of the data sets to figure 5 to the origin. Moreover, erosion may begin as soon as the notch outlet is open, with no tend to a null flux when the time from the opening tends to zero.

Some modification I propose in the conclusion :

paragraph 1 (p.11 l. 2-5) The experiments here suggest that water volume, rather than discharge, [-(remove) controls the total volume of erosion in ravines because sediment discharge rates are linearly related to water discharge rates,] [+(add in place) controls the total volume of erosion during the ravine formation.] This result holds true for both transport-limited and detachment-limited systems.

paragraph 2 (p.11 l. 5-7) As long as slope is a free parameter in these rapidly-evolving systems, [-(remove) each] [+(add in place) changes in] flow rate can be accommodated through [-(remove) changes] [+(add in place) an adjustment] in both cross-sectional and longitudinal channel geometry. Wider channels were typically shorter and thus steeper.

paragraph 3 (p.11 l. 8). "well known" is an unnecessary precision, I would delete it.

I hope those comments could help to improve this article and make a great publication.

Yours sincerely, Dr Valentin Wendling

---

## Referee Comment (RC1) · R. Wells (Referee) · 6 Nov 2017

Manuscript Review: 10.5194/hess-2017-567

Manuscript Title: Impacts of changing hydrology on ravine growth: experimental results

Manuscript Authors: Stephanie S. Day, Karen B. Gran, and Chris Paola

Summary: The manuscript shares details of an experimental study of ravine growth using a saturated substrate and regulated overland flow water volumes. Based on the experimental design, the authors make a case for channel width defined through hydraulic geometry. Sediment transport is modeled in the usual case, with the exception of varying slope, which the authors explain accounts for the linear nature of sediment flux.

[Figure]

**HESSD**

Review: There are several aspects of the study that should be reported that are absent. While the study is very intriguing, an overall lack of experimental methods limits the utility of the results and impact the vitality of the interpretations.

First, simply listing that you used a topographic scanner provides limited information on the applicability of DEMs obtained during overland flow. Perhaps more applicable to the "mud" samples, how exactly was the bed position determined for slope calculations during overland flow? If the entire sample had a sheet flow, then how were elevations determined beneath the flow? Was flow stopped so topography could be determined? Also, please clarify the gridding choice of 2 x 2.5 cm.

Second, there should be more information concerning the soils and their preparation. "Mud" is extremely generic. And, did anyone look at the bulk density of these samples? Water content and bulk density have an impact on erodibility and, therefore, an impact on the outcome of the experiment. Please be explicit concerning the sample, especially how the sample was prepared. Also, what was the initial slope of the samples?

Third, records of this type are not unique, as many have reported similar findings (not referenced). And, why allow slope to vary in your sediment transport model but not in your widening model? Also, comparisons made between rainfall derived erosion and overland flow (only) erosion are not comparable. Why then use Istanbulluoglu for comparisons? Volume is the only comparable term in the two studies, i.e. overland flow volume (current work) and storm volume (past).

As an opinion for the work, I believe that the experimenters neglected important issues of the study in this report. I do not think that the interpretations are incorrect but are skewed to the experiment. The interpretations are eloquent and the overall presentation is very well positioned. However, I regret that, without pertinent information concerning the systems employed and material used, the manuscript should be sent back to the authors for major revision.

I will be happy to review another version.

---

## Referee Comment (RC2) · Anonymous Referee #2 · 27 Dec 2017

This paper deals with the important problem of gully or ravine growth. The paper has its merits as it presents a considerable amount of experimental data (20 experiments in total) using two different substrates and different hydrology settings. Some interesting problems such as width – discharge relation and modeling sediment transport are tack-led. However, in its present form it needs still a bit of work. The results, presentation of the results and analysis is quite confusing. I believe the results are very interesting, but they need to be "fleshed out" considerably before this paper can be published.

**Main comments**

-The introduction gives a very good overview of the problem and state of the art. -The article claims to be studying ravines or permanent gullies. Yet this experiment is clearly tailored at measuring what happens during a newly forming incision, that erodes from

scratch. I believe the results would be different if erosion would be analysed in an existing channel (or existing permanent gully). These might well respond to changes in discharge. -The choice and description of materials is crucial and should be better explained. "Mud" is not an objective description as far as I know. The 96 micrometer substrate classifies as sand following objective classification systems (0.05 - 2 mm)and the finer one as silt. What are the indications for cohesion and that the finer substrate really acts as detachment limited? What is the critical shear stress of these materials? Other properties such as angle of repose will also greatly influence wall stability and could be useful to include. -The experimental setup strikes me as odd to investigate ravine erosion/concentrated flow erosion. Why was a flat bed used? This way the flow does not concentrate until the outlet? From the only picture that is included of the experiments (Figure 1) it seems like you have multiple gully/ravine heads in the mud substrate. What is the effect of this on your results? -The presence of several knickpoints in the mud case seems to indicate the presence of plunge pools. These could potentially be very important yet I am missing an explanation on this. See for example results by Govers et al. Earth Science Reviews 2007. - The presentation of experimental results is quite confusing. The authors use a mix of "water delivery rate", flow rate and discharge (figure 3), sediment volume removed versus mass (it should be easy to convert the first into solid discharge rates using the bulk density of the sediment) etc. Also a mixture of units is used (m3 in the text versus cm3 in figure 3) -It is not surprising to me that no good relation can be found with discharge. Previous studies, mentioned in the introduction, clearly indicate that gullies grow fast in early stages and then reach a more stable state. Figure 4 and 5 clearly shows this as sediment flux peaks in the beginning and then declines. -The text includes a lot of statements that are not backed up by data. For example, p.6 lines 11-12: "channels in the sand erode due to head cut propagation as well as lateral channel migration". No picture or DEM is included however comparing the results and evolution in both substrates so there is no way to check this. The only available figure 6 is difficult to compare and raises further questions. I recommend to align this graphs
in the same direction (flow direction for example, with the outlet facing up). In the mud, two gully heads have formed. This confirms my point made earlier about the experimental setup. How was this handled in the data analysis? As flow splits, how was this modeled? Is the measured channel width the sum of both channels? could this explain why you see no increase in width with increase of discharge for the mud case. See papers by Torri et al. on using channel junctions of rills to model width downstream (10.1016/j.geomorph.2005.11.010). Another example are the statements about slope that are made in the text, yet no discussion is given on the slope results presented in table 2. - In paragraph 3.2 the authors are mixing results and materials and methods or model description. I believe these equations should go in materials and methods. -p8 line 5 "our sediments were easily eroded" Isn't this contradictory to claiming the mud case is detachment limited? -What is the use of describing your experimental conditions in a table if you then cite flow rates etc. in every figure (example fig5)? -p.9 lines 1-4. This is an interesting conclusion. I think it merits some expansion, which comes after line 25. I think this would be clearer if the two paragraphs that are in between were moved. This width-discharge relation is used commonly in gully erosion models but lacks differentiation according to material type as far as I know. Did you measure top width or bottom width? I could not deduct this from the materials and methods. In many field studies bottom width was measured.

**Minor comments**

-Why not use the more common term gully erosion in stead of ravine erosion? (2844 terms in WoS versus 212 results) - if SI units are not deemed illustrative, please use at least standard abbreviations throughout the text for example p5line 1: ml/s figure 2 time (min) not (m) -table 1. Why call this flow? Flow rate or discharge. -The quality of the figures is not very high and looks like they have been done using a standard spread-sheet. Please improve using a more professional graphing programme (R, sigmaplot, grapher, whatever,...). -p.6 lines 1-2 and assuming that you have Dunnean runoff generation as you have in your experiments. With Hortonian overland flow generation, the
whole situation changes. -p.5 line 28. Explain meaning of the threshold line. -p7 line 2 Nachtergaele -figure 7. Please indicate the exact values used for b. 0.4 or 0.39 (text??). What is the value in case a.

---

## Author Comment (AC4) · 24 Jan 2018

[revised manuscript text omitted]

The effects of changing hydrology on newly evolved channels are difficult to study in nature, but physical and numerical models can be used to examine long-term channel evolution under a range of conditions. In one a numerical study, Istanbulluoglu et al. (2005) tested the effect of changing rain intensity while storm volume was held constant. The modeledmodelled results of their study showed an increase in the volume of sediment eroded as intensity increased. This result is in direct conflict with our results, yet there are many distinctions between the two studies that may explain this discrepancy. The Istanbulluoglu et al. (2005) model assumed gully erosion due to slab failure in a detachment-limited system. While the fine grainedmud substrate in our physical experiments was also detachment limited, erosion occurred grain by grain rather than as large blocks. Once these grains were detached, the flow was easily able to carry them through the channel and

there was no measurable deposition in any of the experimental runs. In contrast, erosion in the slab failure model occurred in response to pore pressure build up in tension cracks resulting in large failures. This slab failure model does not require that the flow be able to carry the detached sediment and often resulted in deposition at the toe of the knickpoint, which increases resistance to future failure.

5   ~~Experiments focused on headcut growth completed by Bennett et al. (2000) also reported a linear relationship between water discharge and sediment discharge, yet the water discharge was lower, and the slope of the relationship was much higher in our results. This may suggest that a nonlinear relationship between sediment and water may develop over a wider range of flow rates than tested here, yet more research would be required. In addition, Bennett et al. (2000) noted two dominant processes for head cut migration: surface seal failure, which is described similar to the slab failure mechanism described above, and~~
10  ~~plunge pool scour, where headcut migration is driven by undercutting. While both of these mechanisms could lead to large blocks of sediment collecting at the base of the headcut, creating periods of quiescence in headcut migration as observed by Istanbulluoglu et al. (2005), the authors do not indicate this occurred,  with headcut propagation at a near constant rate after a short period of adjustment. The erosion mechanisms described by Bennett et al. (2000) are similar to the mechanisms observed in our experiments, in that there was a continuous headcut prorogation, yet we didn't observe significant plunge pool~~
15

It is likely that both slab failure and granular knickpoint propagation occur in ravines throughout the world. The relative importance of each process is dependent on the substrate and the knickpoint slope. Tension cracks develop behind steep slopes where shrinkage occurs due to desiccation and horizontal tensile stresses generated in large part by gravity are greater than the tensile strength of the sediment (Darby and Thorne, 1994). Cracks like this are likely to form
20  on the landscape above steep ravine head cuts in cohesive sediment, between storm events. In neither our study nor the Bennett et al. (2000) did we model  individual storm events rather than a constant overland flow. If we had, it is likely we would have also developed tension cracks in the cohesive substrate s. Because this was outside of the scope of these experiments it is difficult to form accurate conclusions on how the development of tension cracks and the subsequent failure events would have altered these results, but an analogous study that encouraged erosion by slab failure would be a
25  useful extension.

30

 mud substrate

5

 modelled

10 ~~relationship with discharge alone. Slope in the Wells et al., (2013) study could not change during the experiment, which contrasts with our experiments where slope is a free parameter that adjusts to discharge. This finding further supports the distinction between rapidly evolving channels like ravines, and more stable systems where slope does not change as quickly causing channel adjustments to occur primarily through changes in channel width and through a non-linear erosion response.~~

Our experimental results are not clear with regard to Sidorchuk's (1999) two-stage ravine evolution model.

15 For most runs it appears that the second stage of ravine evolution was not achieved.– In a few specific cases, most notably, runs F-6 and F-11 there does appear to be an early peak and later decrease and stabilization in sediment discharge. Surprisingly the results of these runs were not among the lowest total sediment discharges, as might be expected where the second stage was achieved.

20 ~~of the experiment. The initial high sediment flux we observed corresponds with the initial set up of the ravine slope and width. The slope and width was then maintained throughout the experiment unless the flow rate was increased corresponding to the steady decrease in sediment flux. When the flow rates increased there was a second increase and rapid decrease in sediment transport rate where the new channel slope and width was formed.~~ It is possible that if these 
[revised manuscript text omitted]

|-----|-------------------------------|---------------|-----------|------------------|--------------------------|--------------------------|
| MF-1 | 0.011920 | 0.17 | 0.06 | 1.19 | 0.22 | |
| MF-2 | 0.024506 | 0.20 | 0.04 | 0.60 | 0.48 | |
| MF-3 | 0.017937 | 0.21 | 0.04 | 0.36 | 0.24 | |
| MF-4 | 0.013776 | 0.27 | 0.03 | 0.62 | 0.43 | |
| MF-5 | 0.010384 | 0.29 | 0.04 | 1.22 | 0.26 | |
| MF-6 | 0.014903 | 0.12 | 0.10 | 0.27 | 0.13 | |
| MF-7 | 0.08913 | 0.42 | 0.11 | 0.71 | 0.38 | |
| MF-8 | 0.013908 | 0.37 | 0.06 | 0.67 | 0.43 | |
| MF-9 | 0.018645 | 0.22 | 0.07 | 0.58 | 0.52 | |
| MF-10 | 0.039053 | 0.15 | 0.05 | 0.81 | 0.40 | 0.49 |
| MF-11 | 0.030158 | 0.14 | 0.05 | 0.56 | 0.16 | 0.28 |
| MF-12 | 0.035532 | 0.16 | 0.03 | 0.29 | 0.38 | 0.61 |
| S-13 | 0.017762 | 0.06 | | | 0.20 | |
| S-14 | 0.018649 | 0.07 | | | 0.13 | |
| S-15 | 0.023796 | 0.06 | | | 0.28 | |
| S-16 | 0.016590 | 0.06 | | | 0.27 | |
| S-17 | 0.023569 | 0.06 | | | 0.29 | |
| S-18 | 0.031174 | 0.07 | | | 0.19 | 0.39 |
| S-19 | 0.030587 | 0.06 | | | 0.14 | 0.43 |
| S-20 | 0.028028 | 0.06 | | | 0.15 | 0.24 |

[Figure]

Constant Head Tank

Constant Volume Tank

Settling Basin

Erodible Substrate

Notch Outlet

**Figure 1: The experimental set up shown here allows water to flow from a settling basin over an erodible substrate and out through a 7.6 x 14 cm notch. The flow rates entering the basin range from 4 to 311 cm$^3$·s$^{-1}$ and are controlled by a constant head tank. For each run a constant volume of water either 190 or 380 litres is run over the erodible substrate. This figure shows the set up for the fine grained  substrate, but the sand substrate set up was similar, yet the erodible substrate was larger.**

[Figure]

**Figure 2: These hydrographs show the range of flow rates tested.** ̶ Each set of curves is for a separate set of experimental runs. ̶ A) 190 litres of water over the  mud substrate. ̶ B) 380 litres of water over the  mud substrate. ̶ C) 190 litres of water over the sand substrate. D) 380 litres of water over the sand substrate.

[Figure]

[Figure]

[Figure]

**Figure 3: The results from these experiments show that the total volume of sediment removed is not dependent of flow rate (A) or that sediment discharge is linearly related to water discharge (B).**  **These charts also show that while sediment discharge is the same for the 380 litre runs as it is for the 190 litre runs at a given discharge, the total volume of sediment removed is greater in the 380 litres runs.**

[Figure]

**Figure 4: Samples collected for 5 seconds every minute during the first 190 liters of run 11 show a decrease in sediment load with time after an initial peak.**

[Figure]

[Figure]

**Figure 5:** In general, sediment discharge was near constant throughout the experiments as indicated by the linear relationship between percent time elapsed and percent of total sediment removed in the graphs above. This result was generally consistent between– for both fine grained substrate (a-d) and the sand substrate (e-h). Results from the 190 L runs are graphed in a and e and

the 380 L runs are graphed in b and f with the first half broken out in c and g and the second half in f and h. The sediment flux over time in the 190 liter runs peaks and later decreases for both mud (A) and sand (B) runs. In the 380 liter runs sediment flux initially peaked for the first 190 liters and then underwent a second peak when the flow was increased for the second 190 liters. This trend can be seen in both the mud (C) and sand (D) runs, where the increase discharge is indicated with a gray marker.

[Figure]

**Change after flow increase (mm)**

- ▊ -130–-39
- ▊ -39–-15
- ▊ -15–15
- ▊ 15–50
- ▊ 50–170

(a)

(b)

**Change after flow increase (mm)**

- ▊ -130–-39
- ▊ -39–-15
- ▊ -15–15
- ▊ 15–50
- ▊ 50–170

(a)

(b)

[Figure]

**Figure 6: These differenced DEMs show how changes in flow impact channel width.** In the **fine grained**  runs (A) the width changed in the newly formed channel, while in the sand runs (B) the channel width was altered along the entire channel length. **Flow is bottom to top in both images.**

[Figure]

Figure 7: Channel width can be modelled using the hydraulic geometry relationship.  In both plots the solid black line is a 1:1 line A) The relationship measured in the channels formed in the **fine grained**  substrate has a constant *b* value that is less than the values commonly observed in nature.  B) The relationship measured in the channels formed in the sand substrate has a constant *b* value of 0.4, which in the intermediate range of the commonly proposed constants.

---

## Author Comment (AC5) · 24 Jan 2018

**Supplementary Materials**

**Run 1:**

[Figure]

**Scan 2:** 15 min & 0.00409 m$^3$ removed          **Scan 3:** 30 min & 0.00657 m$^3$ removed

[Figure]

**Scan 4:** 45 min & 0.00981 m$^3$ removed          **Scan 5:** 60 min & 0.01192 m$^3$ removed

$D_{50}$ = 12 μm

Total Volume = 190 L

Total Run Time = 60 minutes

Flow Rate = 52.58 ml/sec

Total Volume Removed: 0.01192 m$^3$

**Run 2:**

[Figure]

**Scan 2:** 8 min & 0.00585 m$^3$ removed              **Scan 3:** 16 min & 0.01473 m$^3$ removed

[Figure]

**Scan 4:** 24 min & 0.01992 m$^3$ removed              **Scan 5:** 32 min & 0.02451 m$^3$ removed

$D_{50}$ = 12 μm

Total Volume = 190 L

Total Run Time = 32 minutes

Flow Rate = 98.58 ml/sec

Total Volume Removed: 0.02451 m$^3$

**Run 3:**

[Figure]

**Scan 2:** 4 min & 0.00259 m$^3$ removed    **Scan 3:** 8 min & 0.00778 m$^3$ removed

[Figure]

**Scan 4:** 12 min & 0.01.33cm$^3$ removed    **Scan 5:** 16 min & 0.01794 m$^3$ removed

$D_{50}$ = 12 μm

Total Volume = 190 L

Total Run Time = 16 minutes

Flow Rate = 197.16 ml/sec

Total Volume Removed: 0.01794 m$^3$

**Run 4:**

[Figure]

**Scan 2:** 6 min & 0.00404 m$^3$ removed          **Scan 3:** 12 min & 0.00908 m$^3$ removed

[Figure]

**Scan 4:** 18 min & 0.01256 m$^3$ removed          **Scan 5:** 21.25 min & 0.01378 m$^3$ removed

$D_{50}$ = 12 μm

Total Volume = 190 L

Total Run Time = 21.25 minutes

Flow Rate = 148.45 ml/sec

Total Volume Removed: 0.01378 m$^3$

**Run 5:**

[Figure]

[Figure]

**Scan 2:** 4 min & 0.00330 m³ removed          **Scan 3:** 8 min & 0.00735 m³ removed

[Figure]

**Scan 4:** 11 min & 0.01038 m³ removed

$D_{50}$ = 12 μm

Total Volume = 190 L

Total Run Time = 11 minutes

Flow Rate = 286.77 ml/sec

Total Volume Removed: 0.01038 m³

**Run 6:**

[Figure]

[Figure]

**Scan 2:** 30 min & 0.00638 m³ removed          **Scan 3:** 60 min & 0.00901 m³ removed

[Figure]

[Figure]

**Scan 4:** 90 min & 0.01056 m³ removed          **Scan 5:** 120 min & 0.01185 m³ removed

[Figure]

[Figure]

**Scan 6:** 150 min & 0.01255 m³ removed          **Scan 7:** 360 min & 0.01390 m³ removed

[Figure]

**Scan 8:** 738 min & 0.01490 m$^3$ removed      **Scan 9:** 872 min & 0.01490 m$^3$ removed

$D_{50}$ = 12 μm

Total Volume = 190 L

Total Run Time = 872 minutes

Flow Rate = 3.62 ml/sec

Total Volume Removed: 0.01490 m$^3$

**Run 7:**

[Figure]

**Scan 2:** 10.13 min & 0.00891 m$^3$ removed

D$_{50}$ = 12 μm

Total Volume = 190 L

Total Run Time = 10.13 minutes

Flow Rate = 311.40 ml/sec

Total Volume Removed: 0.00891 m$^3$

**Run 8:**

[Figure]

[Figure]

**Scan 2:** 6 min & 0.00310 m$^3$ removed

**Scan 3:** 12 min & 0.00738 m$^3$ removed

[Figure]

**Scan 4:** 21.5 min & 0.01391 m$^3$ removed

$D_{50}$ = 12 μm

Total Volume = 190 L

Total Run Time = 21.5 minutes

Flow Rate = 146.72 ml/sec

Total Volume Removed: 0.01391 m$^3$

**Run 9:**

[Figure]

[Figure]

**Scan 2:** 3 min & 0.00203 m$^3$ removed            **Scan 3:** 6 min & 0.00720 m$^3$ removed

[Figure]

[Figure]

**Scan 4:** 9 min & 0.01139 m$^3$ removed            **Scan 5:** 15.5 min & 0.01865 m$^3$ removed

$D_{50}$ = 12 μm

Total Volume = 190 L

Total Run Time = 15.5 minutes

Flow Rate = 203.52 ml/sec

Total Volume Removed: 0.01865 m$^3$

**Run 10:**

[Figure]

**Scan 2:** 15 min & 0.00688 m³ removed

[Figure]

**Scan 3:** 30 min & 0.01542 m³ removed

[Figure]

**Scan 4:** 44 min & 0.02316 m³ removed

[Figure]

**Scan 5:** 59 min & 0.03200 m$^3$ removed          **Scan 6:** 74 min & 0.03795 m$^3$ removed

[Figure]

**Scan 7:** 91.5 min & 0.03905 m$^3$ removed

$D_{50}$ = 12 μm

Total Volume = 380 L

Total Run Time = 91.5 minutes

1$^{st}$ 190L Flow Rate = 71.69 ml/sec

2$^{nd}$ 190L Flow Rate = 66.41 ml/sec

1$^{st}$ Half Total Volume Removed = 0.02316 m$^3$

Total Volume Removed: 0.03905 m$^3$

**Run 11:**

[Figure]

[Figure]

**Scan 2:** 15 min & 0.01031m³ removed

**Scan 3:** 30 min & 0.01390 m³ removed

[Figure]

**Scan 4:** 43 min & 0.01712 m³ removed

[Figure]

**Scan 5:** 51 min & 0.02218 m$^3$ removed      **Scan 6:** 59 min & 0.02739 m$^3$ removed

[Figure]

**Scan 7:** 65 min & 0.03016 m$^3$ removed

$D_{50}$ = 12 μm

Total Volume = 380 L

Total Run Time = 65 minutes

1$^{st}$ 190L Flow Rate = 73.36 ml/sec

2$^{nd}$ 190L Flow Rate = 143.39 ml/sec

1$^{st}$ Half Total Volume Removed = 0.01712 m$^3$

Total Volume Removed: 0.03016 m$^3$

**Run 12:**

[Figure]

**Scan 2:** 15 min & 0.01096 m$^3$ removed

[Figure]

**Scan 3:** 30 min & 0.01917 m$^3$ removed

[Figure]

**Scan 4:** 41 min & 0.02338 m$^3$ removed

[Figure]

**Scan 5:** 45 min & 0.02665 m$^3$ removed          **Scan 6:** 49 min & 0.03012 m$^3$ removed

[Figure]

**Scan 7:** 53 min & 0.03385 m$^3$ removed          **Scan 8:** 54.5 min & 0.03553 m$^3$ removed

$D_{50}$ = 12 μm

Total Volume = 380 L

Total Run Time = 54.5 minutes

1$^{st}$ 190L Flow Rate = 76.94 ml/sec

2$^{nd}$ 190L Flow Rate = 233.67 ml/sec

1$^{st}$ Half Total Volume Removed = 0.02338 m$^3$

Total Volume Removed: 0.03553 m$^3$

**Run 13:**

[Figure]

[Figure]

**Scan 2:** 6.5 min & 0.01002 m$^3$ removed          **Scan 3:** 25.25 min & 0.01776 m$^3$ removed

D$_{50}$ = 96 μm

Total Volume = 190 L

Total Run Time = 25.25 minutes

Flow Rate = 124.93 ml/sec

Total Volume Removed: 0.01776 m$^3$

**Run 14:**

[Figure]

[Figure]

**Scan 2:** 10 min & 0.00506 m$^3$ removed

**Scan 3:** 20 min & 0.00877 m$^3$ removed

[Figure]

[Figure]

**Scan 4:** 30 min & 0.01067 m$^3$ removed

**Scan 5:** 56.5 min & 0.01865 m$^3$ removed

$D_{50}$ = 96 μm

Total Volume = 190 L

Total Run Time = 56.5 minutes

Flow Rate = 55.83 ml/sec

Total Volume Removed: 0.01865 $m^3$

**Run 15:**

[Figure]

**Scan 2:** 6 min & 0.01496 m$^3$ removed    **Scan 3:** 13.5 min & 0.02380 m$^3$ removed

D$_{50}$ = 96 μm

Total Volume = 190 L

Total Run Time = 13.5 minutes

Flow Rate = 233.67 ml/sec

Total Volume Removed: 0.02380 m$^3$

**Run 16:**

[Figure]

**Scan 2:** 12 min & 0.01659 m$^3$ removed

$D_{50} = 96$ μm

Total Volume = 190 L

Total Run Time = 12 minutes

Flow Rate = 262.88 ml/sec

Total Volume Removed: 0.01659 m$^3$

**Run 17:**

[Figure]

[Figure]

**Scan 2:** 7 min & 0.01459 m³ removed          **Scan 3:** 14min & 0.02007 m³ removed

[Figure]

**Scan 4:** 20.25 min & 0.02357 m³ removed

$D_{50}$ = 96 μm

Total Volume = 190 L

Total Run Time = 20.25 minutes

Flow Rate = 155.78 ml/sec

Total Volume Removed: 0.02357 $m^3$

**Run 18:**

[Figure]

[Figure]

**Scan 2:** 15 min & 0.01221 m³ removed

**Scan 3:** 39 min & 0.01999 m³ removed

[Figure]

[Figure]

**Scan 4:** 45 min & 0.02508 m³ removed

**Scan 5:** 56 min & 0.03117 m³ remove

$D_{50} = 96$ μm

Total Volume = 380 L

Total Run Time = 56 minutes

1st 190L Flow Rate = 80.88 ml/sec

2nd 190L Flow Rate = 185.56 ml/sec

1st Half Total Volume Removed = 0.01999 $m^3$

Total Volume Removed: 0.03117 $m^3$

**Run 19:**

[Figure]

[Figure]

**Scan 2:** 26 min & 0.01109 m$^3$ removed

**Scan 3:** 57 min & 0.01566 m$^3$ removed

[Figure]

[Figure]

**Scan 4:** 64 min & 0.02415 m$^3$ removed

**Scan 5:** 71.5 min & 0.03059 m$^3$ removed

$D_{50}$ = 96 μm

Total Volume = 380 L

Total Run Time = 71.5 minutes

1st 190L Flow Rate = 55.34 ml/sec

2nd 190L Flow Rate = 217.55 ml/sec

1st Half Total Volume Removed = 0.01566 m$^3$

Total Volume Removed: 0.03059 m$^3$

**Run 20**

[Figure]

[Figure]

**Scan 2:** 25 min & 0.00925 m$^3$ removed

**Scan 3:** 50 min & 0.01368 m$^3$ removed

[Figure]

[Figure]

**Scan 4:** 75 min & 0.01587 m$^3$ removed

**Scan 5:** 89 min & 0.02803 m$^3$ remove

$D_{50} = 96 \ \mu m$

Total Volume = 380 L

Total Run Time = 42.06 minutes

$1^{st}$ 190L Flow Rate = 55.34 ml/sec

$2^{nd}$ 190L Flow Rate = 225.32 ml/sec

$1^{st}$ Half Total Volume Removed = 0.01587 $m^3$

Total Volume Removed: 0.02803 $m^3$

---

## Author Response (AR1)

Dear reviewers and editors,

Thank you for all the comments We have worked to improve the manuscript based on these suggestions and feel that the new 5 paper has improved to address many of these concerns. A consistent comment was on the absence of several material measurements for the substrates. Unfortunately, because we no longer have access to these materials, we cannot make these measurements. We do not believe having this information would alter our results or interpretations of our results, but agree that in the future it would be good to have these data for comparisons with other studies. We have tried to address this concern by providing additional information as well as citations for data sheets from the material manufacturers. We have also tried

10 to ensure that the that there was sufficient information in the paper on preparation of the substrate and experimental parameters such that the study could be repeated by future researchers. In addition, we have addressed two terminology concerns including the use of the term permanent gully rather than ravine and the term fine grained rather than mud.

We have also addressed more substantive comments regarding if we reached the second stage of gully evolution and how our results compare with other studies. To address these, we have modified figure 5 significantly and added to our discussion to incorporate other comparable research. We feel this has clarified how our experiments fit into the broader discussion of gully evolution.

1

Below we have included detailed responses to all reviews as well as the marked manuscript.

**20**

Thank you for your contributions to our manuscript.

Sincerely,

25 Stephanie S. Day

Comments on Impacts of Changing Hydrology on Ravine Growth: Experimental Results Hydrol. Earth Syst. Sci. Discuss., https://doi.org/10.5194/hess-2017-567

Dear authors, dear reviewers, dear editors,

I read the experiments with interest. I consider it as an interesting contribution to erosion science. I have a main comment to

5 assess about the interpretation. The total sediment removal volume (Vs) is shown to be not dependent of the flow rate (discharge) (fig 5a). This implies the presented linear relation between the mean water discharge and the mean sediment discharge (fig 5b), as the total volume of water (Vw) is constant, and the total experiment duration (T) is used to calculate both sediment discharge (Qs) and water discharge (Qw): Qs=Vs/T and Qw=Vw/T.

It is also presented that the sediment discharge is not constant during the experiment, at the beginning of each experiment the sediment discharge is high and increases to reach a maximum during the ravine formation, then decreasing to get a significantly

lower value (fig 4 and 5). If the water discharge increases during the experiment, a second peak of sediment discharge is observed following the liquid discharge increase. This is clearly shone on the graphs and explained in the text.

For the longest runs (ie, low water discharge as the water volume is constant in the experiment design) the sediment flux at the end of the experiment seams to have reached a relatively constant value, drastically lower as during the ravine formation (run

15 6 and first stage of run 11 for example). Thus the total removal volume would roughly corresponds to the volume exported during the ravine formation, with a negligible contribution of the flux existing during a following relatively steady state. Indeed if the experiments would be conducted with a higher volume of water delivered at the same discharges, we could expect a quite unchanged removal volume.

Thus it seams confusing to present measured erosion as solid discharge (averaged during the experiment time) and it would be more clear to present it as total removal volume (both in the graph and in the text).

We could consider a first stage during which the ravine forms and adjusts to the constant discharge, with a removal volume not dependent on the discharge, and a second stage where the ravine is in steady state with this discharge. During ravine formation and adjustment to the discharge, the eroded volume would not be dependent on the discharge. During the second stage, the solid discharge is drastically lower and could follow the dependence between solid and liquid discharges known in

25 river systems.

10

20

30

As a second sediment load peak is observed when the discharge is increased (in the experiment conducted with larger water volume – runs 10-12 and 18-20), the total eroded load in a ravine could be function of water discharge fluctuation and temporarily. Note that the total eroded load is higher for experiments conducted with increasing water discharge (and higher water volume) (figure 3a), which could support this hypothesis. Nevertheless, in those experiments the final steady state (if existing) has not been reached (fig 5 c and 5d) mitigating any interpretation.

- Considering the previous comments, I put into question the discussion about the independence between liquid discharge and sediment discharge, it also put into question part of the discussion on the implication of the presented study results for natural systems.
  - 2

Finally I assess some specific comments: As on the precise measurement, the sediment flux is not null at the beginning of the experiment (figure 4), there are no reason to link the first point of the data sets to figure 5 to the origin. Moreover, erosion may begin as soon as the notch outlet is open, with no tend to a null flux when the time from the opening tends to zero.

5 Some modification I propose in the conclusion:

paragraph 1 (p.11 l. 2-5) The experiments here suggest that water volume, rather than discharge, [-(remove) controls the total volume of erosion in ravines because sediment discharge rates are linearly related to water discharge rates,] [+(add in place) controls the total volume of erosion during the ravine formation.] This result holds true for both transport-limited and detachment-limited systems.

10 paragraph 2 (p.11 l. 5-7) As long as slope is a free parameter in these rapidly-evolving systems, [-(remove) each] [+(add in place) changes in] flow rate can be accommodated through [-(remove) changes] [+(add in place) an adjustment] in both cross-sectional and longitudinal channel geometry. Wider channels were typically shorter and thus steeper.

paragraph 3 (p.11 l. 8). "well known" is an unnecessary precision, I would delete it. I hope those comments could help to improve this article and make a great publication.

15 Yours sincerely, Dr Valentin Wendling

Thank you, Dr. Wendling, for this thoughtful and thought-provoking comment. We had considered this issue before, but this got us thinking of better ways to look at the data, resulting in significant modification of figure 5. The figure now shows the sum percent of the sediment removed through time, resulting in a figure that more clearly shows changes to sediment discharge

- 20 through time. We would expect that if we were capturing the second stage of ravine evolution where sediment discharge decreases significantly we would see a non-linear trend, yet this is only seen in a few of the experimental runs. These runs correspond with those runs where the channel interacted with the basin walls. By modifying this figure, we were able to more clearly see that we do not in fact capture the second stage of ravine evolution with a low discharge regime as we originally believed, with a few exceptions. This finding may suggest that the result we have found is only true in the first high discharge
- 25 stage of ravine evolution. This is consistent with our data with the exception of runs 6 and 11 where there was a peak captured and a later decrease in sediment discharge. These runs were consistent in the total volume of sediment removed for the 190 litres, which suggests that perhaps the volume sediment discharge relationship is correct, yet more study would be required to identify this relationship over both stages of ravine evolution.

We have modified the discussion and conclusion to reflect these changes to our thinking in how our results fit into the two

30 stage ravine growth model.

We have also edited the manuscript to reflect all proposed modifications to the conclusion.

Summary: The manuscript shares details of an experimental study of ravine growth using a saturated substrate and regulated overland flow water volumes. Based on the experimental design, the authors make a case for channel width defined through hydraulic geometry. Sediment transport is modeled in the usual case, with the exception of varying slope, which the authors

5 explain accounts for the linear nature of sediment flux.

Review: There are several aspects of the study that should be reported that are absent. While the study is very intriguing, an overall lack of experimental methods limits the utility of the results and impact the vitality of the interpretations.

First, simply listing that you used a topographic scanner provides limited information on the applicability of DEMs obtained during overland flow. Perhaps more applicable to the "mud" samples, how exactly was the bed position determined for slope

- 10 calculations during overland flow? If the entire sample had a sheet flow, then how were elevations determined beneath the flow? Was flow stopped so topography could be determined? Also, please clarify the gridding choice of 2 x 2.5 cm. Second, there should be more information concerning the soils and their preparation. "Mud" is extremely generic. And, did anyone look at the bulk density of these samples? Water content and bulk density have an impact on erodibility and, therefore, an impact on the outcome of the experiment. Please be explicit concerning the sample, especially how the sample was prepared.
- 15 Also, what was the initial slope of the samples?

Third, records of this type are not unique, as many have reported similar findings (not referenced). And, why allow slope to vary in your sediment transport model but not in your widening model? Also, comparisons made between rainfall derived erosion and overland flow (only) erosion are not comparable. Why then use Istanbulluoglu for comparisons? Volume is the only comparable term in the two studies, i.e. overland flow volume (current work) and storm volume (past).

20 As an opinion for the work, I believe that the experimenters neglected important issues of the study in this report. I do not think that the interpretations are incorrect but are skewed to the experiment. The interpretations are eloquent and the overall presentation is very well positioned. However, I regret that, without pertinent information concerning the systems employed and material used, the manuscript should be sent back to the authors for major revision. I will be happy to review another version.

**25**

Thank You, Dr. Wells, for your comments, we have made significant modification to our manuscript to reflect your suggestions, and feel that it has been greatly improved.

Based on your first comment, we have worked to clarify much of our methodology per your suggestion. The laser scanner we used was developed in house at the St. Anthony Falls Laboratory and therefore there is little published information about it.

30 We provided more details about the scanner in page 4 lines 29-31. You also noted the unusual gridding of the scanner. This is a result of the topographic laser scanner system built for these basins. There were two laser scanners that each collected points at 2 cm intervals moving across the basin, each of which were spaced 2.5 cm apart and moved 5 cm after completing a single row of data. The scanner uses a red laser and therefore the flow was indeed stopped to make measurements.

Based on your second comment, information about the substrate was also clarified. The term "mud" was replaced with "fine grained" throughout the text, to more precisely describe the substrate. Unfortunately, we did not collect data regarding the bulk density and water content of each bed preparation and therefore we could not add these data to the text. This would be useful information to record in future studies, and may explain some of the variation was saw, yet we might expect that the variation

5 among the substrate preparations would also reflect the variation within a single substrate, making sampling difficult. Throughout each of our experiments we worked to be as consistent as possible including creating an initially flat bed and vertical knickpoint, yet variability certainly exists even in these controlled settings. This variability is important to note as it is likely a fraction of the variability present in the natural systems we are working to better understand.

Based on your third comment, we have added additional references to our text and modified our discussion to include some additional studies. Regarding your comments on the widening model, we tested the width equation from Wells et al., 2013 that does include slope, but it didn't fit our data as well as the Leopold and Maddock hydraulic geometry equation. A short description about this is included in the discussion section of the paper. We appreciate your comment on the variation in process between rainfall and overland flow, yet in a saturated substrate the erosion should be comparable as no water will

15 our discussion of the Istanbulluoglu paper, because it focuses on volume, which is the focus of this study, and it represents what we anticipated our results to be. Moreover, this paper has a careful discussion of processes observed, and the variation between processes is critical to understanding why our results vary. We did include some additional discussion in page 10 lines 1-10 to reflect this variation in headcut propagation process.

infiltrate the subsurface. While rain splash may contribute to detachment this is generally not significant. Finally, we have kept

This paper deals with the important problem of gully or ravine growth. The paper has its merits as it presents a considerable amount of experimental data (20 experiments in total) using two different substrates and different hydrology settings. Some interesting problems such as width – discharge relation and modeling sediment transport are tackled. However, in its present form it needs still a bit of work. The results, presentation of the results and analysis is quite confusing. I believe the results are

5 very interesting, but they need to be "fleshed out" considerably before this paper can be published. Main comments

Minor comments

- 10 Thank you for your comments on our manuscript. We have made significant modification to the text to reflect your suggestions, and feel that the paper has been greatly improved. Below we have responded to each of your individual comments. -The introduction gives a very good overview of the problem and state of the art. -The article claims to be studying ravines or permanent gullies. Yet this experiment is clearly tailored at measuring what happens during a newly forming incision, that erodes from scratch. I believe the results would be different if erosion would be analysed in an existing channel (or existing permanent gully). These might well respond to changes in discharge.
- permanent gang). These might wen respond to enanges in disenarge.
  - Our results are focused on the development or growth of a large permanent gully or ravine where a steep knickpoint is propagating up onto a flat terrain, they would certainly be different for an existing channel.

-The choice and description of materials is crucial and should be better explained. "Mud" is not an objective description as far as I know. The 96 micrometer substrate classifies as sand following objective classification systems (0.05 - 2 mm) and the

20 finer one as silt.

25

30

 The term "mud" has been replaced with "fine grained" throughout. The materials were selected based on what was available, yet they were effective at capturing two end members of a system.

What are the indications for cohesion and that the finer substrate really acts as detachment limited? What is the critical shear stress of these materials? Other properties such as angle of repose will also greatly influence wall stability and could be useful to include.

 The cohesive substrate was determined to be detachment limited by modeling the ravine growth using detachment limited equations. It is described early as detachment limited which is supported later in the results and discussion sections. This has been further clarified in the text. In addition, the fine grained substrate forms steep vertical walls, which add qualitative support to the idea that the substrate is cohesive and is likely to behave as detachment limited. We agree that these other properties may be useful, but they were not measured in the experiments. In future work this should be considered.

-The experimental setup strikes me as odd to investigate ravine erosion/concentrated flow erosion. Why was a flat bed used? This way the flow does not concentrate until the outlet? From the only picture that is included of the experiments (Figure 1) it seems like you have multiple gully/ravine heads in the mud substrate. What is the effect of this on your results?

- The experiment was originally designed to examine the impacts of hydrology changes on flat agricultural landscapes. We believe that the experiments are interesting without representing a specific case study, so the field area that was the impetus for this study was left out. By only forcing the flow to concentrate at the outlet we are able to investigate the process of head cutting, which is a dominant mechanism for ravine growth.
- 5 -The presence of several knickpoints in the mud case seems to indicate the presence of plunge pools. These could potentially be very important yet I am missing an explanation on this. See for example results by Govers et al. Earth Science Reviews 2007
  - In some cases there was a small secondary gully head that formed, yet a typically there was a larger head that was
    the most dominant. These are an excellent reminder of the natural complexity in a system. Here while we tried to
    control the system closely, there was always quite a bit natural variability. We have included a brief description of
    plunge pools in our discussion. We have elements of plunge pools in our system, yet are missing other elements,
    such as significant undercutting and blocky failures.

10

15

25

30

-The presentation of experimental results is quite confusing. The authors use a mix of "water delivery rate", flow rate and discharge (figure 3), sediment volume removed versus mass (it should be easy to convert the first into solid discharge rates using the bulk density of the sediment) etc. Also a mixture of units is used (m3 in the text versus cm3 in figure 3)

 Discharge and m3 is now being used throughout. The volume vs mass is a result of each graph being presented in the units the data were collected in. Bulk density was not measured for each experiment and therefore the conversion cannot be made.

-It is not surprising to me that no good relation can be found with discharge. Previous studies, mentioned in the introduction,

20 clearly indicate that gullies grow fast in early stages and then reach a more stable state. Figure 4 and 5 clearly shows this as sediment flux peaks in the beginning and then declines.

• While other studies do find a rapid early channel evolution followed by a stable state, they do not explore how discharge impacts that period. We would argue that our data suggest that the length of the rapid growth phase may be dependent on the flow rate. It is also worth noting that it isn't clear from our data that all channels reached that stable state, only in our lowest discharge case did the channel erosion approach zero, in some experiments that sediment discharge continued to increase or was stable throughout the experiment. It might be interesting to extend the experiments to lower volumes of water, I am inclined to believe that the results would be similar.

-The text includes a lot of statements that are not backed up by data. For example, p.6 lines 11-12: "channels in the sand erode due to head cut propagation as well as lateral channel migration". No picture or DEM is included however comparing the results and evolution in both substrates so there is no way to check this. The only available figure 6 is difficult to compare and

- raises further questions. I recommend to align this graphs in the same direction (flow direction for example, with the outlet facing up). In the mud, two gully heads have formed. This confirms my point made earlier about the experimental setup. How was this handled in the data analysis? As flow splits, how was this modeled? Is the measured channel width the sum of both channels? could this explain why you see no increase in width with increase of discharge for the mud case. See papers by Torri
- 35 et al. on using channel junctions of rills to model width downstream (10.1016/j.geomorph.2005.11.010).
  - 7

- The image in figure six shows some lateral channel migration because the channel eroded along the entire length
  and was made wider, but did so only on one side. All images will be made available in a supplement which will
  make this clearer for other experiments as well so the statement is not simply an observation, but is rather backed by
  additional data.
- The graphs have been aligned the same direction.
  - Channel width was measured downstream of the confluence between the multiple channels. This was clarified in
    the text. Channels did widen as discharge was increased, they simply didn't widen downstream of the newly eroded
    areas. We might expect that given enough time to fully adjust to the new discharge that the downstream reaches of
    the gully would have also widened.
- 10 Another example are the statements about slope that are made in the text, yet no discussion is given on the slope results presented in table 2. -In paragraph 3.2 the authors are mixing results and materials and methods or model description. I believe these equations should go in materials and methods.
  - The methods section focuses on the methodology of the experiment to answer our guiding question for this
    research. Additional information is provided as analysis in the results and analysis section of the paper. As these
    equations focus on additional analysis rather than the methods of answering the overall guiding question they are
    not appropriate for that section.

-p8 line 5 "our sediments were easily eroded" Isn't this contradictory to claiming the mud case is detachment limited?

Once detached the sediments were easily removed from the system

-What is the use of describing your experimental conditions in a table if you then cite flow rates etc. in every figure (example **20** fig5)?

**• The graphs were modified to use run name rather than indicate flow rate in the legend.**

-p.9 lines 1-4. This is an interesting conclusion. I think it merits some expansion, which comes after line 25. I think this would be clearer if the two paragraphs that are in between were moved. This width-discharge relation is used commonly in gully erosion models but lacks differentiation according to material type as far as I know. Did you measure top width or bottom width? I could not deduct this from the materials and methods. In many field studies bottom width was measured.

- The paragraphs mentioned were shifted.
- Channel top width was measured, yet in our substrates bottom width is essentially equal to top width. In the fine
  grained substrate this was a result of the sediment's cohesion which formed nearly vertical walls. In the sand
  substrate this is a result of the backs being user low and therefore the bottom and top user impossible to
- 30

25

5

15

substrate this is a result of the banks being very low and therefore the bottom and top were impossible to differentiate.

-Why not use the more common term gully erosion in stead of ravine erosion? (2844 terms in WoS versus 212 results)

**We have replaced the term ravine with permanent gully throughout the manuscript.**

35 - if SI units are not deemed illustrative, please use at least standard abbreviations throughout the text for example p5line 1: ml/s figure 2 time (min) not (m) -table 1. Why call this flow? Flow rate or discharge.

**• Abbreviations have been standardized in figures and text.**

-The quality of the figures is not very high and looks like they have been done using a standard spreadsheet. Please improve using a more professional graphing programme (R, sigmaplot, grapher, whatever, . .).

- Thank you for your feedback
- 5 -p.6 lines 1-2 and assuming that you have Dunnean runoff generation as you have in your experiments. With Hortonian overland flow generation, the whole situation changes.
  - Thank you

-p.5 line 28. Explain meaning of the threshold line. -p7 line 2 Nachtergaele -figure 7. Please indicate the exact values used for

- 10 b. 0.4 or 0.39 (text??). What is the value in case a.
  - The values reported for b in the text and on the figure (0.39 for the sand and 0.27 for the fine grained substrate) are the values that gave the best fit for our data. These are slightly lower than those found for gullies and ravines, yet are within the range for bedrock and alluvial streams.

9

15 • Because the values of a are not typically reported since this is simply a scaling factor we didn't include that in the text. The values we got for a were 16.8 in the sand channel and 8.5 in the fine grained substrate.

**Impacts of Changing Hydrology on **Ravine Permanent Gully** Growth: Experimental Results**

Stephanie S. Day1, Karen B. Gran2, Chris Paola3

[revised manuscript text omitted]

Commonly erosion in detachment-limited systems is modeled modelled by the stream power incision model (Howard and Kerby, 1983):

5
$$dz/dt = kQ^{(m_d)}S^{(n_d)}$$

where k,  $m_d$  and  $n_d$  are constants,  $dz_f dt_k^{-L}$  is vertical erosion rate, and S is the slope. In the detachment-limited fine grainedmud substrate, all erosion took place at the knickpoint and therefore we consider only knickpoint retreat rate and use  $S_k$ , knickpoint slope, in place of S.—Because this rate is a horizontal retreat rate rather than vertical retreat incision rate we must convert Eq. (2) appropriately:

$$\quad U \cdot S_{k} = -dz/dt \tag{3}$$

$$U = kQ^{(m_d)} S_k^{(n_d-I)} \tag{4}$$

where U is the knickpoint retreat rate. The exponents  $m_d$  and  $n_d$  have been derived for a variety of natural environments. Typically the  $m_d \neq n_{a_b} \frac{1}{r}$  ratio (concavity index) is approximately 0.35 - 0.6 (Whipple and Tucker, 1999; Baldwin et al., 2003). The exponent  $n_d$  ranges between 2/3 and 5/3 depending on the erodibility of the substrate where more easily eroded sediment has a lower value for the exponent  $n_d$  (Foley 1980; Howard and Kirby 1983; Whipple et al., 2000).— The exponents  $m_d$  and  $n_d$

for uniform detachment-limited landscapes (i.e. badlands) reported by Howard and Kerby (1983) are 4/9 and 2/3, respectively. While the  $m_d \neq n_{d_s} \frac{1}{L}$  ratio is slightly higher than reported elsewhere, we used these values to model erosion in our experiments because, like the badlands, our experimental set-up was spatially homogeneous and easily eroded.—RMSE was measured between the calculated U and the measured U and minimized by modifying the coefficient k. -The RMSE was 25-% of the

20 measured average knickpoint retreat rate for the fine grained substrate.— The detachment-limited equation is not appropriate for the sand substrate because there is both headward and lateral erosion, which is not captured by the equation. In addition, the knickpoint slope cannot be measured in the sand substrate.

Sediment loads from the sand substrate were modeled modelled with the transport-limited equation (Pelletier, 2011):

$$q_s = kQ^{(m_t)} S^{(n_t)}_{-}$$

15

25 where k,  $m_t$  and  $n_t$  are constants and  $q_s$  is the volumetric unit sediment flux.—[This equation can be derived from the Engelund and Hansen (1967) equation where both  $m_t$  and  $n_t$  equal 5/3. Engelund and Hansen (1967) is the ideal equation to use because it does not have an incipient motion threshold as many other sediment transport equations do. Here again the RMSE was

17

(2)

(5)

| Formatted: Font: Not Bold |  |
|---------------------------|--|
| Formatted: Font: Not Bold |  |

measured between the measured and calculated  $q_s$  and minimized using the coefficient *k*.—The RMSE was 41-% of the average volumetric unit sediment discharge for the sand substrate.—The transport-limited equation was also tested with the fine grained substrate. For this test the average slope was used for  $S_s$  and the RMSE calculated was 86% of the average volumetric unit sediment discharge. This high RMSE value supports the observation that the fine grained substrate does not behave as a transport limited system.

**4 Discussion**

5

30

Our experiments demonstrate that, over a range of conditions, the sediment volume eroded during ravine-permanent gully growth under application of a fixed volume of water is independent of the rate at which the water is supplied. Thus, sediment discharge is linearly related to water discharge in both detachment-limited and transport-limited systems. These results contrast with data from many pre-existing streams where changing flow intensity has resulted in increased erosion volume (iei.e., Boateng et al., 2012; Ma et al., 2010; Naik and Jay, 2011), yet this previously observed responsethese data may not be directly applicable to early-stage ravinesgullies.— We suggest that because ravines-gullies are actively evolving in response to a given hydrology, the channel morphology that develops reflects that hydrology, with erosion balanced by altering channel slope.— This is supported by sediment transport Eqs. (4) and (5); which, when applied to our experiments, are able to predict the measured sediment discharges by including the effects of both discharge and slope.— In pre-existing channels where channel slope may take tens of thousands of years to adjust to changing flows, both of these equations would predict a non-linear increase in sediment discharge with increasing water discharge.— Ravines Permanent-Ggullies\_evolve more rapidly in response to the imposed discharge and can balance erosion by adjusting channel slope in response to a change in the hydrologic regime.

20 Moreover our findings suggest that anthropogenic changes to discharge regime could affect channel morphology (i.e. channel width), without changing sediment input-output derived from ravinespermanent gullies after an initial response period.—It is important to note that there was an initial increase in sediment discharge when Where water discharge was increased in the 3890 literlitre runs, but the channel quickly evolved in response to the new discharge by creating a wider channel. In these experiments the observed response to the increased discharge differed for the fine grainedmud and substrate (Fig. 6) suggesting cohesion may be a dominant n important factor in the how a channel initially responds to a new discharge regime.

The results of our experiments follow the hydraulic geometry relationship for width in Eq. (1) although with lower exponents than usually measured in field studies for alluvial and bedrock channels (Knighton, 1998; Leopold and Maddock, 1953; Montgomery and Gran, 2001; Whipple, 2004) and ephemeral-rills and ephemeral gullies (Nachtergaele et al., 2002; Torri et al., 2006). While this relationship is typically applied to describing width or discharge changes in a single channel, it works here for these separate channels because each comparable channel is carving through the same substrate. In the fine grained

substrate the empirical exponent b for these data is lower than has been derived for natural channels. This may be a result of the steep channel walls developed in these experiments; in natural **ravinpermanent gullies** where near vertical channel walls are less common, we expect that this exponent would be closer to the reported values. In the sand substrate where steep channel walls could not develop, the empirical exponent b is 0.4039 which is only slightly lower than the range typically considered

5 for alluvial channels (Rodriquez-Iturbe and Rinaldo, 1997), but similar to exponents found in rills and ephemeral gullies (Nachtergaele et al., 2002; Torri et al., 2006).

Channel width has also been modelled by Wells et al. (2013) who found that both slope and discharge play a role in setting channel width. This relationship was tested for the results of these experiments, yet it was not as strong as the relationship with discharge alone. Slope in the Wells et al. (2013) study could not change during the experiment, which contrasts with our
experiments where slope is a free parameter that adjusts to discharge. This finding further supports the distinction between rapidly evolving channels like ravinepermanent gullies, and more stable systems where slope does not change as quickly causing channel adjustments to occur primarily through changes in channel width and through a non-linear erosion response.

Experiments focused on headcut growth completed by Bennett et al. (2000) also reported a linear relationship between water
discharge and sediment discharge, yet the water discharge was lower and the slope of the relationship was much higher in our results. This may suggest that a nonlinear relationship between sediment and water may develop over a wider range of flow rates than tested here, yet more research would be required. In addition, Bennett et al. (2000) noted two dominant processes for head cut migration: surface seal failure, which is similar to slab failure reported in other papers, and plunge pool scour, where headcut migration is driven by undercutting. Although both of these mechanisms could lead to large blocks of sediment
collecting at the base of the headcut, creating periods of quiescence in headcut migration, the authors do not indicate that headcut migration stalled in their experiments. The erosion mechanisms described by Bennett et al. (2000) are similar to the mechanisms observed in our experiments.

The effects of changing hydrology on newly evolved channels are difficult to study in nature, but physical and numerical models can be used to examine long term channel evolution under a range of conditions. In one-a\_numerical study, Istanbulluoglu et al. (2005) tested the effect of changing rain intensity while storm volume was held constant.—\_The modeledmodelled results of their study showed an increase in the volume of sediment eroded as intensity increased. -This result is in direct conflict with our results, yet there are many distinctions between the two studies that may explain this discrepancy.-The Istanbulluoglu et al. (2005) model assumed gully erosion due to slab failure in a detachment-limited system.

rather than as large blocks.—Once these grains were detached, the flow was easily able to carry them through the channel and there was no measurable deposition in any of the experimental runs.—In contrast, erosion in the slab failure model occurred in response to pore pressure build up in tension cracks resulting in large failures.—This slab failure model does not require that the flow be able to carry the detached sediment and often resulted in deposition at the toe of the knickpoint, which increases resistance to future failure.

Experiments focused on headcut growth completed by Bennett et al. (2000) also reported a linear relationship between water discharge and sediment discharge, yet the water discharge was lower, and the slope of the relationship was much higher in our results. This may suggest that a nonlinear relationship between sediment and water may develop over a wider range of flow rates than tested here, yet more research would be required. In addition, Bennett et al. (2000) noted two dominant processes
for head cut migration: surface seal failure, which is described similar to the slab failure mechanism described above, and plunge pool scour, where headcut migration is driven by undercutting. While both of these mechanisms could lead to large blocks of sediment collecting at the base of the headcut, creating periods of quiescence in headcut migration as observed by Istanbulluoglu et al. (2005), the authors do not indicate this occurred, with headcut propagation at a near constant rate after a short period of adjustment. The erosion mechanisms described by Bennett et al. (2000) are similar to the mechanisms observed
in our experiments, in that there was a continuous headcut prorogation, yet we didn't observe significant plunge pool

**development.**

20

25

30

5

It is likely that both slab failure and grain by graingranular knickpoint propagation occur in ravines permanent gullies throughout the world.— The relative importance of each process is dependent on sediment typethe substrate and the knickpoint slope. —Tension cracks develop behind steep slopes where shrinkage occurs due to desiccation and horizontal tensile stresses generated in large part by gravity are greater than the tensile strength of the sediment (Darby and Thorne, 1994). Cracks like this are likely to form on the landscape above steep ravine head cuts in cohesive sediment, between storm events.—If weln neither our study nor the Bennett et al. (2000) did we model-had modeled individual storm events rather than a constant overland flow. If we had, it is likely we would have also developed tension cracks in the cohesive mud substrates.—Because this was outside of the scope of these experiments it is difficult to form accurate conclusions on how the development of tension cracks and the subsequent failure events would have altered our these results, but an analogous study that encouraged erosion by slab failure would be a useful extension.

The results of our experiments follow the hydraulic geometry relationship for width in Eq. (1) although with lower exponents than usually measured in field studies for alluvial and bedrock channels (Knighton, 1998; Leopold and Maddock, 1953; Montgomery and Gran, 2001; Whipple, 2004) and ephemeral rills and gullies (Nachtergacle et al., 2002; Torri et al., 2006).

While this relationship is typically applied to describing width or discharge changes in a single channel, it works here for these separate channels because each comparable channel is carving through the same substrate. In the fine grainedmud substrate the empirical exponent *b* for these data is lower than has been derived for natural channels. This may be a result of the steep channel walls developed in these experiments; in natural ravines where near vertical channel walls are less common, we expect that this exponent would be closer to the reported values. In the sand substrate where steep channel walls could not develop, the empirical exponent *b* is 0.40 which is only slightly lower than the range typically considered for alluvial channels (Rodriquez Iturbe and Rinaldo, 1997), but similar to exponents found in rills and ephemeral gullies (Nachtergaele et al., 2002; Torri et al., 2006).

5

Channel width has also been modeledmodelled by Wells et al., (2013) who found that both slope and discharge play a role in setting channel width. This relationship was tested for the results of these experiments, yet it was not as strong as the relationship with discharge alone. Slope in the Wells et al., (2013) study could not change during the experiment, which contrasts with our experiments where slope is a free parameter that adjusts to discharge. This finding further supports the distinction between rapidly evolving channels like ravines, and more stable systems where slope does not change as quickly causing channel adjustments to occur primarily through changes in channel width and through a non-linear erosion response.

- 15 Our experimental results also lend support toare not clear with regard to Sidorchuk's (1999) two-stage ravine-gully evolution model. For most runs it appears that the second stage of gully evolution was not achieved.— In a few specific cases, most notably, runs F-6 and F-11 there does appear to be an early peak and later decrease and stabilization in sediment discharge. Surprisingly the results of these runs were not among the lowest total sediment discharges, as might be expected where the second stage was achieved. While it is not clear that the ravines we produced reached the second state stage of ravine evolution,
- 20 they did show initially high sediment transport rates which decreased rapidly and then showed a slow steady decrease through the rest of the experiment. The initial high sediment flux we observed corresponds with the initial set up of the ravine slope and width. The slope and width was then maintained throughout the experiment unless the flow rate was increased corresponding to the steady decrease in sediment flux. When the flow rates increased there was a second increase and rapid decrease in sediment transport rate where the new channel slope and width was formed. It is possible that if theseall the runs
- 25 were allowed to continue we may have reached a stable system where the second stage of gully evolution was achieved. If this was allowed to occur, we would anticipate that additional water at the same discharge would not cause measurable erosion, potentially altering the relationship we have observed.- What is not clear is how long it takes for this steady state to occur, and how this may relate to discharge.

Based on the results of this study and the comparison with previous ravine gully studies, it is important to consider a wide ange of variables when mitigating ravine permanent gully erosion.—\_The results from our experiments suggest that during early stages of ravine permanent gully growth, increasing overland flow rates will not result in increased sediment yield, if the

volumes of water delivered are not changed.-\_Moreover, while sediment loads are not affected by changing flow rates, channel morphology is.-\_ Another important variable to consider is the mode of head cut retreat.-\_The experimental results apply in environments with steep slopes where erosion is grain-by-grain. In places where tension cracks develop, the slab failure mechanism highlighted by Istanbulluoglu et al. (2005) might dominate.-\_If the tension crack failure mechanism is the dominant process of head cut retreat, flow rates and storm intensity may become more important than they were in our study because the slabs may require a higher threshold to break up and mobilize, allowing further head-cut propagation.

**5** Conclusion**

5

10

15

20

25

These experiments highlight how young incising channels like ravines-permanent gullies can respond to changing hydrology differently than higher order channels that are later in their evolution. A relevant future study should investigate how natural ravines-gullies, which have a great deal more variability than this natural experimental system, respond to changing hydrology. The conclusions from this project are outlined below:

- The experiments here suggest that water volume, rather than discharge, controls the total volume of erosion in ravines because sediment discharge rates are linearly related to water discharge ratesduring permanent gully formation.— This result holds true for both transport-limited and detachment-limited systems.
- As long as slope is a free parameter in these rapidly-evolving systems, changes inchanges in flow rate can be accommodated through changes an adjustment in both cross-sectional and longitudinal channel geometry. Wider
- In both substrates, variations in channel width were described by the well-known hydraulic geometry relationship proposed by Leopold (1953), with wider channels forming in response to higher discharge.
  - Sediment transport in sand and fine grainedmud substrates was well described by the transport-limited sediment flux equation and the detachment-limited stream power equation, respectively.
  - Sediment transport was greatest at the beginning of each run and slowed through time, increasing again after discharge was increased. The high sediment transport likely corresponds with the time when the channel width and slope were being set.

[revised manuscript text omitted]

I

| Table 1: Experimental R | Run Parameters |
|-------------------------|----------------|
|-------------------------|----------------|

| Run                   | Substrate | Water    | Total                 | Flow 1st   | Flow 2 nd |  |
|-----------------------|-----------|----------|-----------------------|------------|----------------------|--|
|                       |           | Volume   | Time                  | 190 liters | 190 liters           |  |
|                       |           | (liters) | (min) $(cm^3 s^{-1})$ |            | $(cm^3 s^{-1})$      |  |
| MF -1          | Fine      | 190      | 60.0                  | 52.58      |                      |  |
|                       | Mud       |          |                       |            |                      |  |
| ₩ F -2  | MudFine   | 190      | 32.0                  | 98.58      |                      |  |
| MF -3          | MudFine   | 190      | 16.0                  | 197.16     |                      |  |
| MF -4          | MudFine   | 190      | 21.25                 | 148.45     |                      |  |
| MF -5          | MudFine   | 190      | 11.0                  | 286.77     |                      |  |
| MF -6          | MudFine   | 190      | 872.0                 | 3.62       |                      |  |
| MF -7          | MudFine   | 190      | 10.13                 | 311.40     |                      |  |
| MF -8          | FineMud   | 190      | 21.5                  | 146.72     |                      |  |
| MF -9          | FineMud   | 190      | 15.5                  | 203.52     |                      |  |
| MF -10         | FineMud   | 380      | 91.5                  | 71.69      | 66.41                |  |
| MF -11         | FineMud   | 380      | 65.0                  | 73.36      | 143.39               |  |
| ₩ F -12 | FineMud   | 380      | 54.5                  | 76.94      | 233.67               |  |
| S-13                  | Sand      | 190      | 25.25                 | 124.93     |                      |  |
| S-14                  | Sand      | 190      | 56.5                  | 55.83      |                      |  |
| S-15                  | Sand      | 190      | 13.5                  | 233.67     |                      |  |
| S-16                  | Sand      | 190      | 12.0                  | 262.88     |                      |  |
| S-17                  | Sand      | 190      | 20.25                 | 155.78     |                      |  |
| S-18                  | Sand      | 380      | 56.0                  | 80.88      | 185.56               |  |
| S-19                  | Sand      | 380      | 71.5                  | 55.34      | 217.55               |  |
| S-20                  | Sand      | 380      | 89.0                  | 42.06      | 225.32               |  |

| Run                  | Volume Sediment           | Average | Bed Slope | Knickpoint | Width 1st 190 | Width 2nd 190 |
|----------------------|---------------------------|---------|-----------|------------|---------------|---------------|
|                      | Removed (m 3 ) | Slope   |           | Slope      | liters (m)    | liters (m)    |
| MF -1         | 0.011920                  | 0.17    | 0.06      | 1.19       | 0.22          |               |
| MF -2         | 0.024506                  | 0.20    | 0.04      | 0.60       | 0.48          |               |
| MF -3         | 0.017937                  | 0.21    | 0.04      | 0.36       | 0.24          |               |
| MF -4         | 0.013776                  | 0.27    | 0.03      | 0.62       | 0.43          |               |
| ₩ F -5 | 0.010384                  | 0.29    | 0.04      | 1.22       | 0.26          |               |
| MF -6         | 0.014903                  | 0.12    | 0.10      | 0.27       | 0.13          |               |
| MF -7         | 0.08913                   | 0.42    | 0.11      | 0.71       | 0.38          |               |
| MF -8         | 0.013908                  | 0.37    | 0.06      | 0.67       | 0.43          |               |
| MF -9         | 0.018645                  | 0.22    | 0.07      | 0.58       | 0.52          |               |
| MF -10        | 0.039053                  | 0.15    | 0.05      | 0.81       | 0.40          | 0.49          |
| MF -11        | 0.030158                  | 0.14    | 0.05      | 0.56       | 0.16          | 0.28          |
| MF -12        | 0.035532                  | 0.16    | 0.03      | 0.29       | 0.38          | 0.61          |
| S-13                 | 0.017762                  | 0.06    |           |            | 0.20          |               |
| S-14                 | 0.018649                  | 0.07    |           |            | 0.13          |               |
| S-15                 | 0.023796                  | 0.06    |           |            | 0.28          |               |
| S-16                 | 0.016590                  | 0.06    |           |            | 0.27          |               |
| S-17                 | 0.023569                  | 0.06    |           |            | 0.29          |               |
| S-18                 | 0.031174                  | 0.07    |           |            | 0.19          | 0.39          |
| S-19                 | 0.030587                  | 0.06    |           |            | 0.14          | 0.43          |
| S-20                 | 0.028028                  | 0.06    |           |            | 0.15          | 0.24          |

5

Figure 1: The experimental set up shown here allows water to flow from a settling basin over an erodible substrate and out through a 7.6 x 14 cm notch.—The flow rates entering the basin range from 4 to 311 cm3s-1 and are controlled by a constant head tank.—For each run a constant volume of water either 190 or 380 literslitres is run over the erodible substrate.—This figure shows the set up for the fine grained mud substrate, but the sand substrate set up was similarsimilar, yet the erodible substrate was larger.

---

## Author Response (AR2)

We appreciate all the comments from Reviewer 3. Below we have outlined how we adjusted our manuscript considering these suggestions. In addition, we answer the reviewer's questions.

Thank you

Stephanie S. Day

We have added in the reference you suggested to add context for the study and note that the experimental work could span the time and space scales of rills (covered by Govers et al., 2007) to permanent gullies or ravines.

10    We agree that shear stress should have been measured, but without a measurement of flow depth or velocity this was impossible to calculate. It would be an interesting extension to measure these variables as well. Despite this shortcoming this work sheds light on how erosion in an actively evolving may vary simply as a result of changing flow rates. As mentioned it is known that shear stress is important in erosion yet shear stress can be varied by altering flow in several ways by altering channel width or changing flow velocity for instance. By allowing our channel to freely evolve the shear stress is allowed to adjust to the

15    discharge by altering the channel to match.

Corrected "rate of delivery" or "water delivery rate" to flow rate. We do use both the term "flow rate" and "discharge" interchangeably. In general, flow rate is used when we are referring specifically to the experimental approach, while discharge is used more generally when comparing our results to theoretical models of incision and erosion. We did update the units to ensure that these were consistent throughout the manuscript.

20    We worked to make figure 5 clearer by enlarging and darkening text.

Equation reference was added to figure 7

Both substrates are artificial, they were selected to ensure that we had reduced complexity for these experiments so the focus would be on the gully growth process rather than adding in an additional complication that may come into effect with variable grain sizes. The technical data for each of the substrates is cited in the paper (Zeespheres G series data sheet & AGSCO

25    technical data), and could be inspected for more details on the sediments. While it is true that natural systems exist with greater grain size variability this paper focuses on a single process or one part of the natural system. In a natural system complexity is certainly increased not just with grain size variability but also due to vegetation, changes in flow throughout a storm hydrograph, differential substrate moisture, and many other variables that are beyond the scope of this work. Experiments are still relevant in that they provide an opportunity to better understand simplified processes that contribute to our understanding

30    of natural systems.

We feel our work adds to the body of literature related to channel formation and fluvial geomorphic process. In addition, it clearly and concisely demonstrates that increased overland flow rates will not lead to increased rates of erosion if water volume is unchanged.

[revised manuscript text omitted]